



# Coral reef-derived dimethyl sulfide and the climatic impact of the loss of coral reefs

Sonya L. Fiddes[1,2,3,*], Matthew T. Woodhouse[3], Todd P. Lane[4], and Robyn Schofield[4]

[1]Australian-German Climate and Energy College, University of Melbourne, Parkville, Australia
[2]ARC Centre of Excellence for Climate System Science, School of Earth Sciences, University of Melbourne, Parkville, Australia
[3]Climate Science Centre, Oceans and Atmosphere, Commonwealth Scientific and Industrial Research Organisation, Aspendale, Australia
[4]ARC Centre of Excellence for Climate Extremes, School of Earth Sciences, University of Melbourne, Parkville, Australia
[*]Now at the Australian Antarctic Program Partnership, Institute of Marine and Antarctic Science, University of Tasmania, Hobart, Australia

**Correspondence:** S. Fiddes (sonya.fiddes@utas.edu.au)

**Abstract.** Dimethyl sulfide (DMS) is a naturally occurring aerosol precursor gas which plays an important role in the global sulfur budget, aerosol formation and climate. While DMS is produced predominantly by phytoplankton, recent observational literature has suggested that corals and their symbionts produce a comparable amount of DMS, which is unaccounted for in models. It has further been hypothesised that the coral reef source of DMS may modulate regional climate. This hypothesis

presents a particular concern given the current threat to coral reefs under anthropogenic climate change. In this paper, a global climate model with online chemistry and aerosol is used to explore the influence of coral reef-derived DMS on atmospheric composition and climate. A simple representation of coral reef-derived DMS is developed and added to a common DMS surface water climatology, resulting in an additional DMS flux of $0.3 \, \mathrm{Tg \, year^{-1}}$ S, or 1.7% of the global flux. By comparing the differences between both nudged and free running ensemble simulations with and without coral reef-derived DMS, the

influence of coral reef-derived DMS on regional climate is quantified. In the Maritime Continent-Australian region, where the highest density of coral reefs exist, a small decrease in nucleation and Aitken mode aerosol number concentration and mass is found when coral reef DMS emissions are removed from the system. However, these small responses are found to have no robust effect on regional climate via direct and indirect aerosol effects. This work emphasises the complexities of the aerosol-climate system and the limitations of current modelling capabilities are highlighted, in particular surrounding

convective responses to changes in aerosol. In conclusion we find no robust evidence that coral reef-derived DMS influences global and regional climate.



## 1 Introduction

Marine organisms (phytoplankton, algae) are known to produce the chemical dimethyl sulfoniopropionate (DMSP). In the
ocean, DMSP experiences enzymatic cleavage forming dimethyl sulfide (DMS, Yoch, 2002, see Figure 1 point 1), which can
then be released into the atmosphere (2). Atmospheric DMS ($DMS_a$) can undergo a series of chemical reactions to become
a sulfate aerosol (3). When in sufficiently large abundance (4) these sulfate aerosol can impact aerosol loading and cloud
properties, altering the radiation budget directly (5) and indirectly (6) and having a cooling effect (7). This effect has been
hypothesised to form a short-term bioregulatory negative feedback system, known as the CLAW (Charlson, Lovelock, Andreae
and Warren) hypothesis (Charlson et al., 1987), whereby marine organisms can alter their environment when stressed. This
hypothesis remains unproven, and arguments against it cite the complexity and non-linearity of the DMS-climate system (Quinn
and Bates, 2011; Green and Hatton, 2014). Nevertheless, at longer time scales, global modelling studies have shown that marine
derived DMS plays an important role in maintaining the current large-scale climate (Thomas et al., 2010; Woodhouse et al.,
2010; Gabric et al., 2013; Mahajan et al., 2015), providing a global cooling effect of up to $0.45\,°C$ (Fiddes et al., 2018). Many
global DMS-climate modelling studies have also considered DMS under future scenarios (Bopp et al., 2004; Gabric et al.,
2004; Kloster et al., 2007; Cameron-Smith et al., 2011; Six et al., 2013; Grandey and Wang, 2015; Schwinger et al., 2017).
However, considering our understanding of DMS in the current climate remains uncertain, the aforementioned studies do not
provide a clear consensus on how DMS production may respond to a warming climate. With this in mind, better knowledge of
current sources of DMS is important to further our understanding of DMS-climate interaction now and into the future.

One such source of DMS that is currently unaccounted for in climate modelling is coral reefs. Recent studies have shown
that corals, coral symbionts, and coral by-products (eg. mucus) produce large amounts of DMSP (Broadbent et al., 2002;
Broadbent and Jones, 2004; Jones and Trevena, 2005; Jones et al., 2007; Burdett et al., 2015; Jackson et al., 2020a), especially
when stressed. Of note for this work, Jones et al. (2018) has summarised reports of $flux_{DMS}$ values of $0\text{-}4906\,\mu g\,m^{-2}\,day^{-1}$
and a mean of $205\,\mu g\,m^{-2}\,day^{-1}$ in summer and $0.6\text{-}481\,\mu g\,m^{-2}\,day^{-1}$ with a mean of $77\,\mu g\,m^{-2}\,day^{-1}$ over winter over the
Great Barrier Reef (GBR). Jones et al. (2018) also suggest that total emissions from the GBR are equivalent to $0.02\,Tg\,year^{-1}$
of sulfur. This estimate has been made from measurements both over coral reefs and in the GBR lagoon, and also includes an
estimate of the additional flux from tropical cyclones. The tropical cyclone emission has been calculated (not observed) using
the Liss and Merlivat (1986) flux parameterisation, taking into account average wind speeds of tropical cyclones and accounting
for approximately five cyclone days per year in the region. However, it is noted that many parameterisations overestimate DMS
flux ($flux_{DMS}$) at high wind speeds.

In addition, recent work has shown a sensitivity of DMS production by corals when stressed due to tidal exposure, warming
temperatures, rainfall events and light exposure (Swan et al., 2012; Fischer and Jones, 2012; Hopkins et al., 2016; Swan et al.,
2017). Of interest to this study are the findings from Hopkins et al. (2016), where the effect of tidal exposure on the coral species
*Acropora cf.horrida* was studied in laboratory experiments. From their results, Hopkins et al. (2016) extrapolate a $flux_{DMS}$ of
$9\text{-}35\,\mu mol\,m^{-2}\,day^{-1}$ over coral reefs. This estimate is equivalent to $288.6\text{-}1122.3\,\mu g\,m^{-2}\,day^{-1}$ of sulfur, and in this work
is further extrapolated to global coral reef coverage, giving $0.024\text{-}0.12\,Tg\,year^{-1}$ of sulfur. Whilst these extrapolations are





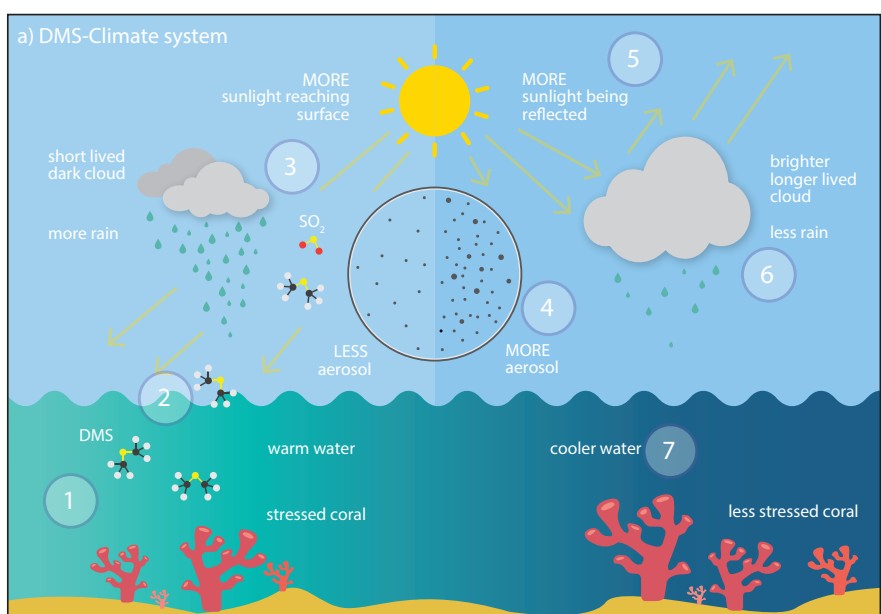

**Figure 1.** Schematic diagram describing the DMS-climate system. The numbers are described in detail in the text. Graphics designed by G. Harmer

highly speculative in terms of estimated exposure time, coverage of coral reefs, account for just one species of coral and only DMS produced during tidal stress, the Hopkins et al. (2016) estimations were the first to attempt to quantify the large-scale flux of coral reef-derived DMS.

Following these observed results, numerous studies have made links to coral DMS, aerosol formation, cloud cover and/or sea surface temperatures (SSTs) (Modini et al., 2009; Deschaseaux et al., 2012; Leahy et al., 2013; Swan et al., 2017; Jones et al., 2017; Cropp et al., 2018; Jackson et al., 2018, 2020b). Jones (2013), Jones et al. (2017) and Cropp et al. (2018) further suggest that coral reefs participate in bioregulatory feedback as suggested by the CLAW hypotheses. Most of these studies do not explicitly account for the complexity of the DMS-climate system and its significant non-linearities (see Thomas et al., 2011;

Quinn and Bates, 2011; Green and Hatton, 2014; Fiddes et al., 2018). In addition, deducing a climatic impact of one aerosol species using observations alone is fraught with co-varying and confounding influences from other aerosol species. These complexities can only be addressed through modelling studies, however no modelling study has included coral reef-derived DMS as a source of sulfur to date.

   To add urgency to this problem, coral reef ecosystems globally are facing dire risk due to anthropogenic climate change

(Hughes et al., 2017, 2018). The IPCC special report on climate change (IPCC, 2018) states that under $1.5\,°C$ warming, 70-90% of coral reefs will be extinct. The risk to coral reefs is two fold; increasing sea surface temperatures are causing more frequent mass coral bleaching events (Hughes et al., 2017; King et al., 2017), whilst increasing ocean acidification is causing





reduced calcification and growth of coral species (Hoegh-Guldberg et al., 2017; Magnan et al., 2016). Whilst the death of global coral reefs due to anthropogenic climate change will have clear impacts on the ecosystems (Plaisance et al., 2011) and
economies (Deloitte Access Economics, 2017; Spalding et al., 2017) they support, a lesser known, and as of yet unquantified impact is upon the climate itself.

In light of these recent studies and the current threat to coral reefs, we raise the questions: what is the influence of coral reef-derived DMS on climate, including its influence on aerosol production and cloud formation, and what implications does mass coral extinction have for the climate? To address these questions we use a global climate-chemistry model to investigate
whether coral reef-derived DMS has an impact on climate. This is the first study to estimate what implications mass coral extinction may have for global and regional climate.

## 2 Methods

### 2.1 ACCESS-UKCA description

The coupled climate-chemistry model ACCESS (Australian Community Climate and Earth System Simulator) - UKCA (United
Kingdom Chemistry and Aerosol) is used in this work to quantify the importance of global coral reef-derived sulfur. The ACCESS-UKCA physical atmospheric model is based on the Global Atmosphere 4.0 configuration of the Unified Model at Version 8.4 (Walters et al., 2014) and the UKCA model includes the state of the art aerosol parameterisation GLObal Model of Aerosol Processes (GLOMAP)-mode scheme (Mann et al., 2010, 2012). Anthropogenic emissions, pre- and post-2000 respectively are provided by Lamarque et al. (2010); van Vuuren et al. (2011) and biomass emissions by van der Werf et al.
(2017). Emissions of other species (biogenic, primary aerosol) are described in Woodhouse et al. (2015). SSTs and sea ice are prescribed following the Atmospheric Model Intercomparison Project (AMIP) method (Taylor et al., 2015). $DMS_w$ is adapted from the Lana et al. (2011) climatology, and is discussed further below, and the Liss and Merlivat (1986) flux parameterisation is used. The UKCA is coupled to the ACCESS model via the radiation scheme and the large-scale cloud and precipitation schemes; both the direct and indirect aerosol forcing are modelled. ACCESS-UKCA has a resolution of $1.25°$ latitude x$1.875°$
longitude with 85 vertical levels. Where the model is nudged, ERA-Interim (Dee et al., 2011) is used at six hourly intervals (via horizontal wind components and potential temperature, see Section 2.3 for more details). Further model details and evaluation are available in Fiddes et al. (2018).

### 2.2 DMS climatologies

We have developed a DMS surface water concentration ($DMS_w$) climatology based on Lana et al. (2011), in which additional
$DMS_w$ over coral reef regions is included. To determine the amount of $DMS_w$ to be added to the Lana et al. (2011) climatology, the fraction of each ACCESS-UKCA grid box covered by global warm water coral reefs was calculated using the UNEP-WCMC et al. (2010) data. With this areal distribution, a weighted concentration of $DMS_w$ was added to the Lana et al. (2011) $DMS_w$ climatology.





The weighted addition of up to 50 nM (for 100% coverage) of $DMS_w$ caused a global mean increase of 0.03 nM and an

additional $flux_{DMS}$ of 0.3 Tg year$^{-1}$ S. The $flux_{DMS}$ of the (Lana et al., 2011) and the additional coral reef $flux_{DMS}$ are shown in Figure 2. These increases do not impact the global sulfur budget, contributing only 1.7% of additional sulfur to the global $flux_{DMS}$. The 50 nM climatology adds a mean of 0.74 nM and a maximum of 7.8 nM, to coral reef regions, values that are within those found in the literature. The additional daily $flux_{DMS}$ simulated by ACCESS-UKCA over coral reefs, shown in Figure 2b (maximum of 621.9 $\mu$g m$^{-2}$ day$^{-1}$ S), is similar to that of the Hopkins et al. (2016) estimations of $flux_{DMS}$ due

to one coral species in response to tidal stress (288.6-1122.3 $\mu$g m$^{-2}$ day$^{-1}$ S). Furthermore, Jones et al. (2018) suggest that the total $flux_{DMS}$ from the GBR and surrounding lagoon is approximately 0.002 Tg year$^{-1}$ S. If this number is extrapolated to global coral reef regions then an annual $flux_{DMS}$ of 0.12 Tg year$^{-1}$ S is estimated. The values from the Hopkins et al. (2016) and Jones et al. (2018) both suggest that the amount of $DMS_w$ attributed to coral reefs in this thesis is within the high end of what is currently observed.

However, it is noted that the values stated here are averages over large grid-boxes, and so likely overestimate the extent of coral reef influence. Nevertheless, the 50 nM perturbation was chosen in part to ensure that if no significant changes in the atmosphere were found it would not be because the additional coral reef DMS was too small.

## 2.3 Experiment set-up

To study the impact of coral reef-derived DMS in ACCESS-UKCA two sets of simulations were performed: nudged and free-

running. In both sets of simulations a control (using the Lana et al. (2011) climatology, referred to as L11) and experimental simulation (using the Lana et al. (2011) climatology with additional coral and here on refereed to as the L11C50 simulation) were completed. The nudged simulations follow the methods described by Fiddes et al. (2018) in which the control and coral simulations were nudged to the ERA-Interim data set (Dee et al., 2011) in the free troposphere at six hourly intervals using horizontal winds and potential temperature. In the second setup the model was allowed to freely run, with no nudging applied.

In both sets of simulations, SSTs are prescribed. Without nudging, feedbacks from the meteorology, such as changes in wind fields, are able to manifest within the model. In the free-running simulations, much greater model variability manifested and differentiating between a true signal from the perturbed DMS field and internal model variability was difficult. For this reason, seven simulations (for both the L11 and L11C50), each of ten years, were performed with the free running setup to provide an ensemble. Each set of simulations used different atmospheric initial conditions from a previous nudged simulation, taken

at 0000UTC, January 1 for 1996, 1998, 1999, 2000, 2001, 2002 and 2003 (noting that 1997 was excluded as it was a strong El Niño year). Whilst this exercise was computationally expensive it was able to provide sufficient data to perform statistical analyses.

## 2.4 Statistical analysis

The following regions were defined for statistical analysis: the Maritime Continent - Australian (MC-Aus) region from 17.4°S

to 10°N, 95.625°E to 153.75°E; the Queensland (QLD) land only region from 30°S to 10°S, 138.75°E to 153.75°E and the Australian land only region from 45°S to 10°S, 112.5°E to 153.75°E. Four grid points were selected for analysis of aerosol





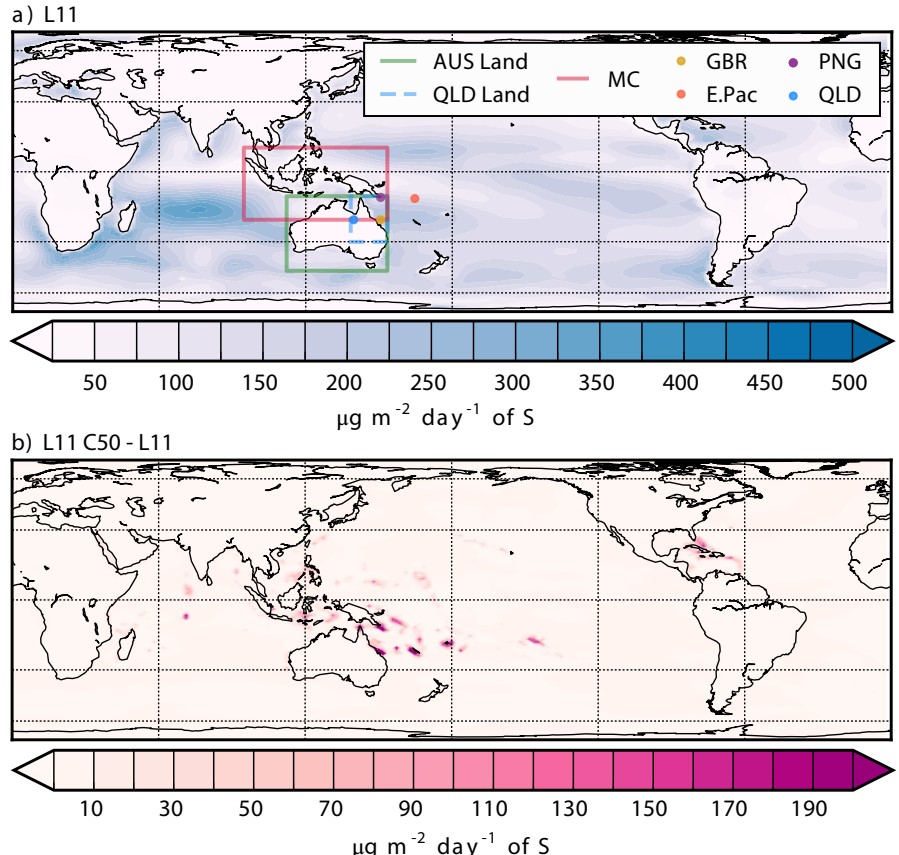

**Figure 2.** a) the flux$_{DMS}$ (in $\mu$g m$^{-2}$ day$^{-1}$ S) based on the Lana et al. (2011) DMS$_w$ climatology and b) the additional flux$_{DMS}$ in the climatology with added coral reef derived DMS$_w$. In a) the Maritime Continent-Australian region shown by a red box, Australian land only region by a green box, Queensland (QLD) land only region by a blue box. Additionally, four single points used for analysis are shown over the Great Barrier Reef (GBR - yellow), Papua New Guinea (PNG - pink), the eastern Pacific (E.Pac- orange) and QLD (blue)

size distribution: a location off the coast of Papua New Guinea (PNG) at 10.5°S, 151°E, a point in the East Pacific (E.Pac) at 10.5°S, 165°E, a location over the GBR at 20°S, 151°E and a point in inland QLD at 20°S, 140°E. These regions and points are shown in Figure 2a. In addition the boundary of the South Pacific Convergence Zone (SPCZ) is defined as the area within
the 6 mm day$^{-1}$ or greater precipitation threshold (Vincent et al., 2011).

To test the significance of differences in a given field, the two-tailed Student T-test (Wilks, 2011) and the respective field significance (Wilks, 2011) are used. Both of these methods have been evaluated at confidence levels of the 95th percentile ($p < 0.05$) unless otherwise stated. In addition to this, ensemble agreement, where at least five out of the seven ensemble members agree on the change in sign, has been shown as stippling as a further indicator of confidence.





## 3 Impacts of coral reef-derived DMS over the MC-Aus region

### 3.1 DMS and sulfur dioxide

This Section describes the changes in atmospheric DMS ($DMS_a$) when coral reefs are removed (L11 minus L11C50), noting that the results are presented this way in order to demonstrate what the impact of the loss of coral reefs may be on the climate system. The annual differences in $DMS_a$ (with respect to the L11C50 simulation, shown in Figure 3a) for the nudged simulations (Figure 3b) and the free ensemble (Figure 3c) are spatially very similar to each other and to the respective change in $flux_{DMS}$ (shown in Figure 1c). Good agreement across the ensemble is found with the free running simulations over reef regions. Figure 3d-g shows, for both the free and nudged simulations, a strong seasonal signal over the MC-Aus region in terms of both mean value and the range of values. A seasonal signal is found in both the $DMS_a$ and $flux_{DMS}$ (see Table 1), and is in part due to the variation of the L11 $DMS_w$ climatology. In DJF, the range of results from the ensemble are larger than those of any other season and it appears that the free running meteorology is actually dampening the response of $DMS_a$ compared to the nudged runs. This damped response is not surprising as surface wind speeds were found to be stronger in this region at this time in the free ensemble (not shown). Increased wind speeds in the free ensemble cause an increased $flux_{DMS}$ leading to a smaller decrease in $flux_{DMS}$ and $DMS_a$ than in the nudged run. Such examples of wind driven DMS responses are also found in other locations (and in other seasons), including over the Southern Ocean at around 60°E and 60°W (see annual plots in Figure 3c). These responses highlight the complexity of the DMS-climate system. In addition, wind driven responses in other aerosol sources are found in regions around the globe (eg. from sea salt and dust), but are not discussed further here.

Figure 4 shows changes in $SO_2$ that are spatially similar to that of $DMS_a$. However the reductions around smaller coral reefs, such as those in the central Pacific, Indian or Caribbean oceans, are of lesser magnitude to that of the coral reefs in the MC-Aus region. This is true for both the nudged (Figure 4b) and the free simulations (Figure 4c) which implies that these areas of lower reef density are unlikely to have a significant impact on regional climate. In terms of seasonality, Table 1 and Figure 4d-g indicate less $SO_2$ variability compared to $DMS_a$ throughout the year for both the free and nudged simulations. MAM (March, April, May) has the smallest, statistically insignificant change (-0.9% and -1.3% for the free and nudged simulations respectively), while DJF (December, January, February) and JJA (June, July, August) show larger changes over the MC-Aus region of -3.2 and -3.8% ($p < 0.05$) in the free ensemble and -2.7 and -2.2% ($p > 0.1$) for the nudged runs. The violin plots indicate there is a similar degree of variability for each of these seasons (compared to $DMS_a$) and the influence of free running meteorology is again noted where changes in $SO_2$ are found outside of coral reef regions.

Interestingly, in the nudged (b) and, to a lesser degree, the free (c) simulations in Figure 4, a small increase in $SO_2$ is found to the east of PNG. This anomaly is larger in both MAM and SON (September, October, November) than in JJA and is almost non-existent in DJF (not shown). The localised increase is found to have an impact on the larger (accumulation size) aerosol number concentrations and the sulfate mass, subsequently increasing the aerosol optical depth (AOD). A similar anomaly is noted in the results presented in Fiddes et al. (2018) and Woodhouse et al. (2019) - a study on secondary organics using the same version of ACCESS-UKCA. These similarities indicate that this anomaly is not specific to coral reef DMS, but is instead





**Table 1.** Seasonal changes over the Maritime Continent-Australian region as a percent (except for $SW\downarrow_{Surf,CS}$, $SW\uparrow_{TOA,CS}$ and $SW\downarrow_{Surf}$ which are an absolute change in $W\,m^{-2}$ and w in $cm\,s^{-1}$) for both the free running ensemble (F) and the nudged simulations (N). Number density is abbreviated as ND, and mass to MS for each of the four aerosol modes (nucleation, Aitken, accumulation and coarse)

| Field | Run | DJF | MAM | | JJA | SON |
|---|---|---|---|---|---|---|
| $Flux_{DMS}$ | F | $-5.1^{95}$ | $-13.7^{95}$ | | $-19.2^{95}$ | $-13.5^{95}$ |
| (%) | N | $-8.8^{95}$ | $-12.2^{95}$ | | $-18.6^{95}$ | $-11.4^{95}$ |
| $DMS_a$ | F | $-6.1^{95}$ | $-14.2^{95}$ | | $-22.3^{95}$ | $-14.3^{95}$ |
| (%) | N | $-8.9^{95}$ | $-12.6^{95}$ | | $-20.6^{95}$ | $-13.3^{95}$ |
| $SO_2$ | F | $-3.2^{95}$ | $-0.9$ | | $-3.8^{95}$ | $-2.4^{95}$ |
| (%) | N | $-2.7$ | $-1.3$ | | $-2.2$ | $-1.5$ |
| $N_3$ | F | $-0.1$ | $-4.6^{95}$ | | $-5.2^{95}$ | $-5.3^{95}$ |
| (%) | N | $-3.3$ | $-3.8^{90}$ | | $-5.1$ | $-3.6^{90}$ |
| Nuc. ND | F | $-0.2$ | $-5.2^{95}$ | | $-6.0^{95}$ | $-6.2^{95}$ |
| (%) | N | $-3.8$ | $-4.4^{90}$ | | $-6.1$ | $-4.6^{95}$ |
| Nuc. MS | F | $-1.4$ | $-6.4^{95}$ | | $-6.5^{95}$ | $-6.6^{95}$ |
| (%) | N | $-4.3$ | $-5.8^{95}$ | | $-5.5$ | $-4.3^{95}$ |
| Ait. ND | F | $0.0$ | $-1.9^{95}$ | | $-2.1^{95}$ | $-1.5$ |
| (%) | N | $-1.3$ | $-1.2$ | | $-2.3^{95}$ | $-1.1$ |
| Ait. MS | F | $-1.6$ | $0.0$ | | $-3.1^{95}$ | $-2.0$ |
| (%) | N | $-2.4$ | $-1.2$ | | $-1.3$ | $-1.2$ |
| Acc. ND | F | $-1.4$ | $0.1$ | | $-1.8$ | $-1.3$ |
| (%) | N | $-0.5$ | $-0.8$ | | $-0.7$ | $-0.7$ |
| Acc. MS | F | $-4.3^{90}$ | $1.1$ | | $-4.3^{95}$ | $-2.0$ |
| (%) | N | $-1.9$ | $-1.5$ | | $-1.1$ | $-0.6$ |
| Coa. ND | F | $1.8$ | $-1.3$ | | $0.0$ | $-1.9$ |
| (%) | N | $-0.4$ | $0.2$ | | $0.2$ | $-0.4$ |
| Coa. MS | F | $-2.3^{95}$ | $-0.7$ | | $-3.1^{95}$ | $-3.4^{95}$ |
| (%) | N | $-0.9$ | $-1.8$ | | $-0.4$ | $-0.7$ |
| $CCN_{70}$ | F | $-0.7$ | $0.3$ | | $-1.4^{90}$ | $-1.4$ |
| (%) | N | $-0.6$ | $-0.8$ | | $-0.9$ | $-0.9$ |

[95] Statistical significant at the 95th percentile, $p < 0.05$
[90] Statistical significant at the 90th percentile, $p < 0.1$





**Table 1.** Seasonal changes over the Maritime Continent-Australian region as a percent (except for $SW\downarrow_{Surf,CS}$, $SW\uparrow_{TOA,CS}$ and $SW\downarrow_{Surf}$ which are an absolute change in $W\,m^{-2}$ and w in $cm\,s^{-1}$) for both the free running ensemble (F) and the nudged simulations (N). Number density is abbreviated as ND, and mass to MS for each of the four aerosol modes (nucleation, Aitken, accumulation and coarse (continued)

| Field | Run | DJF | MAM | JJA | SON |
|---|---|---|---|---|---|
| AOD | F | -1.4 | 0.0 | -1.6[90] | -1.7 |
| (%) | N | -1.1 | -1.1 | -0.6 | -0.6 |
| $SW\uparrow_{TOA,CS}$ | F | -0.04 | 0.00 | -0.06[90] | -0.11[90] |
| ($W\,m^{-2}$) | N | -0.05 | -0.04 | -0.03 | -0.04 |
| $SW\downarrow_{Surf,CS}$ | F | 0.12 | 0.00 | 0.11 | 0.03 |
| ($W\,m^{-2}$) | N | 0.07 | 0.06 | 0.01 | 0.05 |
| CDN | F | -0.6 | 0.1 | -0.9[90] | -0.9 |
| (%) | N | -0.4 | -0.6 | -0.4 | -0.6 |
| LWP | F | 0.3 | -0.8 | 0.4 | 0.4 |
| (%) | N | -0.1 | -0.2 | 0.3 | 0.2 |
| Water Vapour | F | -1.0 | 0.0 | -0.1 | 1.0 |
| (%) | N | 0.1 | 0.1 | -0.1 | -0.1 |
| Low cloud frac. | F | 0.7 | -0.6 | 0.9 | 1.7 |
| (%) | N | 0.1 | -0.1 | -0.4 | 0.2 |
| High cloud frac. | F | -1.0 | -0.6 | 0.5 | -0.1 |
| (%) | N | -0.1 | 0.0 | -0.1 | -0.1 |
| Precip. | F | -0.3 | -0.4 | 0.2 | 0.8 |
| (%) | N | -0.2 | -0.2 | 0.4 | 0.2 |
| Large-scale precip. | F | 0.9 | -0.7 | -0.8 | 1.0 |
| (%) | N | -0.2 | 0.6 | 0.6 | 0.1 |
| Convective precip. | F | -0.4 | -0.3 | 0.3 | 0.8 |
| (%) | N | -0.2 | -0.3 | 0.4 | 0.2 |
| $w_{500hPa}$ | F | -0.009 | -0.004 | -0.001 | 0.015 |
| ($cm\,s^{-1}$) | N | -0.001 | -0.001 | 0.003 | -0.001 |
| $SW\downarrow_{Surf}$ | F | 0.56 | 0.23 | 0.01 | -0.14 |
| ($W\,m^{-2}$) | N | 0.11 | 0.25 | 0.01 | 0.09 |

[95] Statistical significant at the 95th percentile, $p < 0.05$
[90] Statistical significant at the 90th percentile, $p < 0.1$

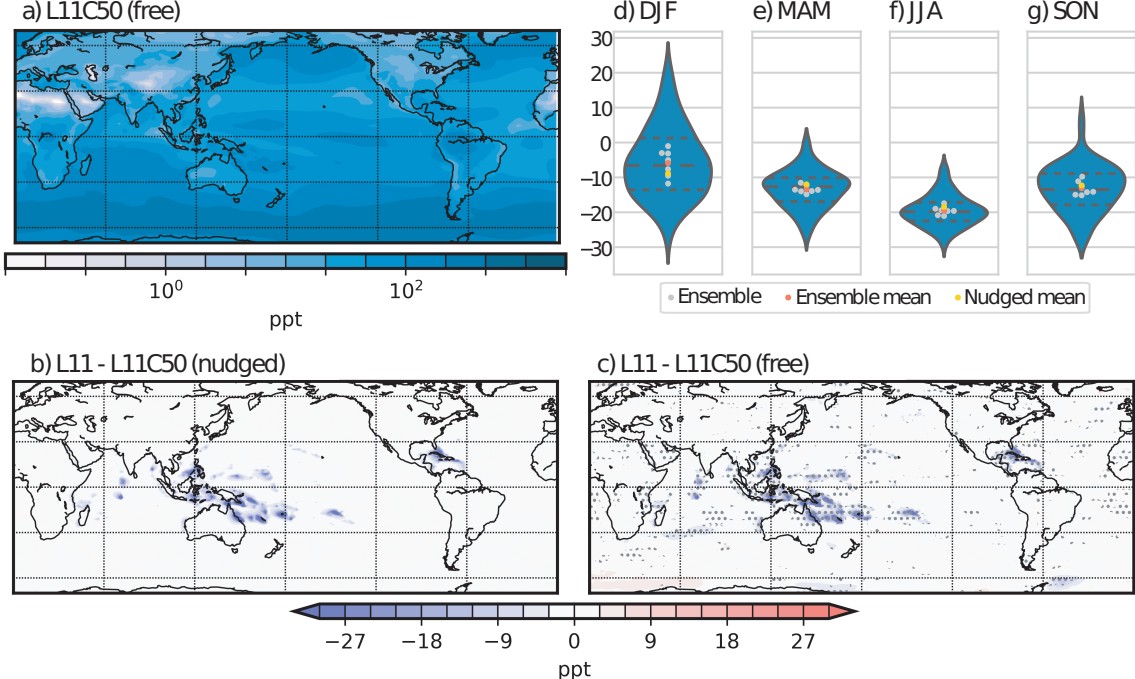

**Figure 3.** The annual $DMS_a$ concentration (ppt) for the L11C50 simulation (a) and the L11-L11C50 difference for the nudged simulations (b) and free running ensemble (c). In c) the model agreement, where at least five of the seven ensemble pairs agree in the sign of the difference, is shown by stippling. d-g) show violin plots of the average seasonal difference (L11-L11C50) in $DMS_a$ over the MC-Aus region as a percentage, where all years in the ensemble is shown by the distribution, the dashed lines represent the 25, 50 and 75th percentiles, the grey dots show each pair of models mean difference, red represents the ensemble average and yellow represents the nudged average

a sensitivity within the model in this region to changes in secondary aerosol. This anomaly has some impact on the results in this area and will be kept in mind for the remainder of this study.

## 3.2 Nucleation and Aitken mode aerosol

The removal of coral reef-derived DMS leads to a significant ($p < 0.05$) decline in nucleation mode aerosol number concentration of -5.1, -6.0 and -6.2% for MAM, JJA and SON respectively in the free ensemble over the MC-Aus region (see Table 1). In the nudged simulations, nucleation mode number concentration decreases by -4.6% ($p < 0.05$) in SON and by -4.4% ($p < 0.1$) in MAM. Although not statistically significant, JJA continues to show the largest changes (-6.1%), and DJF the smallest (-3.8%), in the nudged runs. The changes in nucleation mode number concentration are strongly reflected in the number concentrations of particles greater than 3 nm dry diameter ($N_3$) which are shown annually in Figure 5. The violin plots





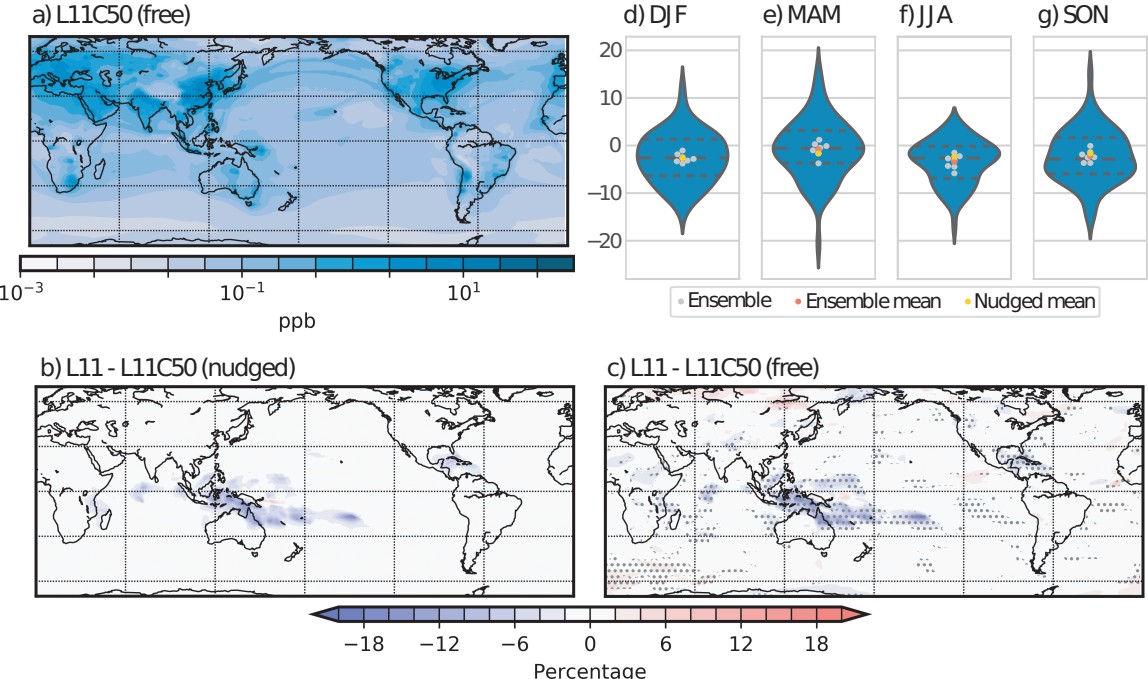

**Figure 4.** As for Figure 3 but for $SO_2$ in parts per billion (ppb) (a) and percent (b-g)

in Figure 5d-g show a large range in the response in DJF, while a much smaller range in SON, with many of the individual free running pairs agreeing on the magnitude of the change.

These small aerosol (nucleation mode sized) show a greater response to changes in DMS than the larger aerosol sizes (see Table 1) in both the free and nudged simulations, aligning with the results of Fiddes et al. (2018). This result indicates that fewer homogeneous nucleation events are occurring in the free troposphere as a result of removing coral reef-derived DMS. Hence, fewer particles are being entrained back into the boundary layer. This process is confirmed by the vertical profile of $N_3$ showing a decrease in the upper levels of the atmosphere (not shown).

The size distributions shown in Figure 6 at four grid points also show the larger decreases are occurring in the nucleation mode and the Aitken mode. The range of locations shown by the size distribution, including directly over coral reefs (GBR and PNG) and removed from coral reefs (East Pacific and QLD) demonstrate that the effect of coral reef-derived DMS loss is not restricted to directly over coral reef regions. This is shown notably at the QLD location for JJA in both the nucleation and Aitken modes and SON for the Aitken mode. Over the MC-Aus region, decreases in the soluble Aitken aerosol number occur in MAM and JJA of -1.9 and -2.1% respectively ($p < 0.05$). The changes in both the nucleation and Aitken mode number concentrations are accompanied by similar changes in aerosol mass at this size (see Table 1).



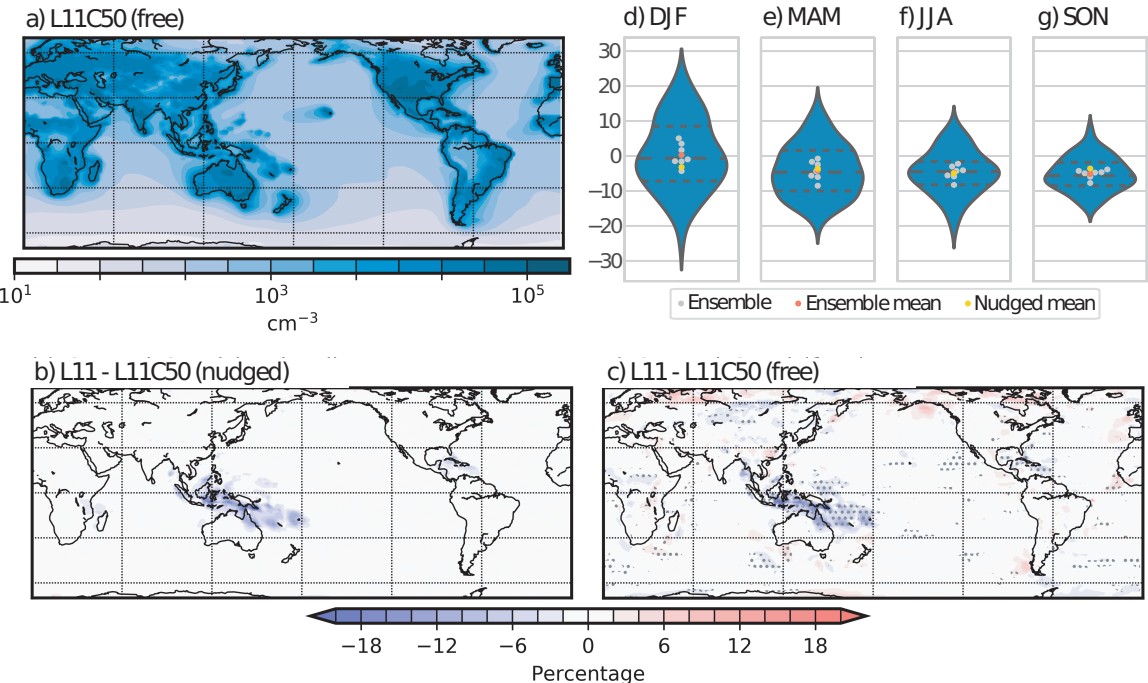

**Figure 5.** As for Figure 3 but for $N_3$ number concentration in $cm^{-3}$ (a) and percent (b-g)

### 3.3 Accumulation and coarse mode aerosol

At the larger sizes (soluble accumulation and coarse modes) little change in aerosol number is found on average over the MC-Aus region. However, significant ($p < 0.05$) decreases in aerosol mass are found in the coarse mode during DJF, JJA and SON, and in the accumulation mode in DJF ($p < 0.1$) and JJA ($p < 0.05$) for the free running ensemble (see Table 1). Declines of smaller magnitude ($p > 0.1$) are found for the nudged runs. For the free ensemble, the larger change in mass instead of number suggests a reduction in aerosol growth at these sizes as the larger aerosol sizes rely on condensational growth and cloud processing to interact with DMS-derived sulfate. One exception to the observed change in mass instead of number is the region west of PNG, which varies by season (not shown). In this region, a small increase in accumulation mode aerosol number is found, accompanied by a larger increase in the aerosol mass. This increase is more evident in the nudged runs and is linked to the sensitivity of the model to $SO_2$ in this region. A similar result can be found in the simulations presented in Fiddes et al. (2018). Subsequently, it is suggested that this positive anomaly is not an effect of coral reef removal and is instead masking some of the effects of removing coral reef-derived DMS when results are averaged across the MC-Aus region, especially in MAM and SON.



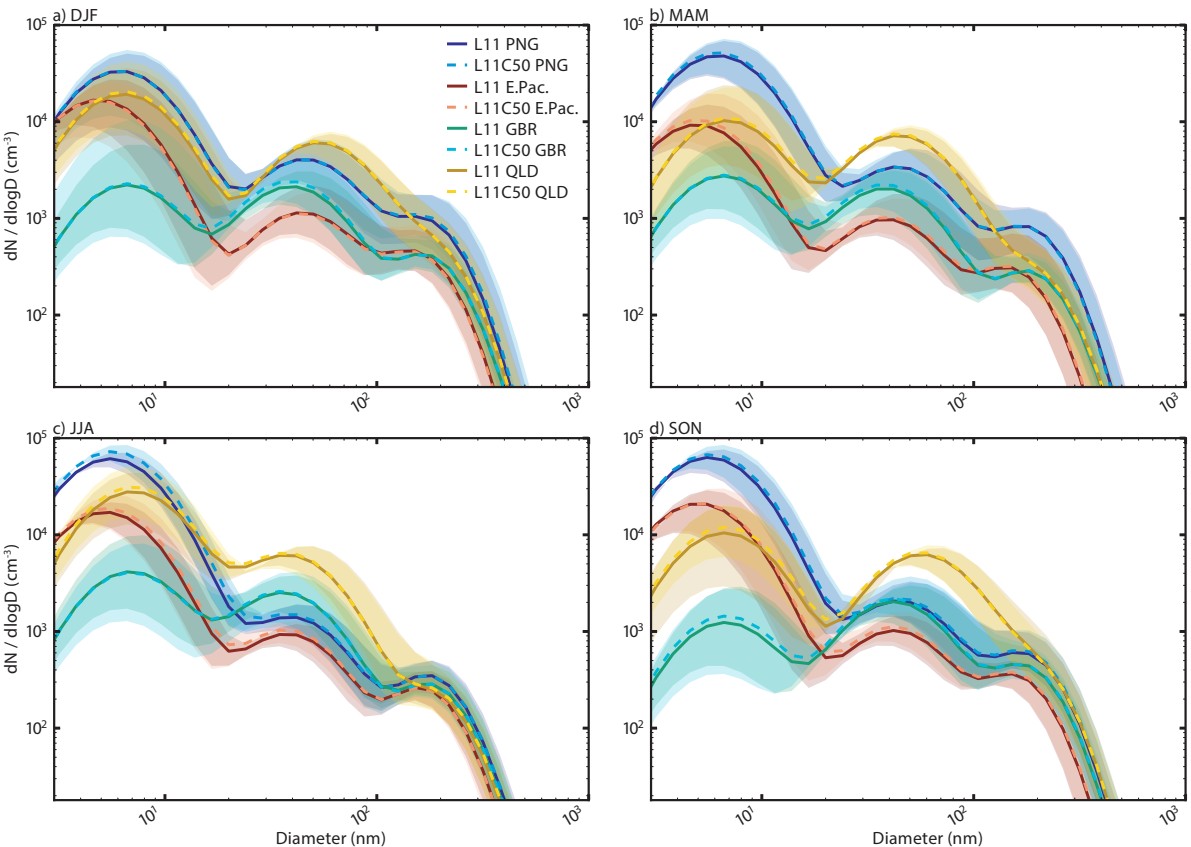

**Figure 6.** Aerosol dry diameter in nm (x-axis) and number concentration (as a function of number over the log of the diameter) in $cm^{-3}$ (y-axis), both in log scales, for four regions: off the coast of PNG (blues), in the East Pacific (reds/oranges), over the GBR (green/turquoise) and inland QLD (yellow/brown), for the L11C50 (dashed lines) and L11 simulations (solid lines) (see Figure 1b for regions), for four season DJF (a), MAM (b), JJA (c), SON (d). The lines show the ensemble mean for the free running simulations and the shaded regions show the range of results from all years in the ensemble

So far we show that the largest response in aerosol due to loss of coral reef-derived DMS occurs over the MC-Aus region. For
this reason, all subsequent plots focus on the MC-Aus region. It is noted that there are impacts on aerosol from the free running meteorology outside of this domain, but these can mostly be explained by changes in surface wind speeds due to variability in the free-running ensemble (not shown). In addition, the influence of the free running meteorology is found to have a larger impact on the results in subsequent analysis, with greater seasonal variation.





### 3.4 Cloud condensation nuclei

Despite the removal of coral reef-derived aerosol having the largest impact on the nucleation mode aerosol, these small aerosol don't interact with model radiation via direct or indirect aerosol effects. While the changes are small in the larger sized aerosol number and mass, a cumulative response to the loss of coral reef-derived sulfur has some interesting impacts. Figure 7 shows the column integrated seasonal cloud condensation nuclei with a dry diameter greater than $70\,\mathrm{nm}$ ($CCN_{70}$) response to removal of coral reef-derived DMS. For the nudged simulations in Figure 7b, f, j and n, a consistent, yet small and insignificant reduction

in $CCN_{70}$ is found over the MC-Aus region (between -0.6 to -0.9%). Interestingly, over the SPCZ region, decreases of -1.0, -0.3, -1.9 and -1.1% are also found in the nudged simulations for each season (DJF, MAM, JJA and SON) respectively. The SPCZ is a relatively clean region, with few $CCN_{70}$ sized aerosol (see Figure 7a) and also few coral reefs. This lack of direct aerosol source suggests that changes in aerosol from other regions is affecting this region via transport along the SPCZ.

For the free ensemble, in Figure 7c, g, k, and o, the SPCZ region again stands out in MAM, JJA and SON with decreases in

$CCN_{70}$ of -2.8, -2.2 and -1.9% for each season respectively (MAM and SON, $p < 0.1$, and JJA, $p < 0.05$). In this region, little change in surface wind speeds are found, suggesting that the changes found here are also likely due to loss of aerosol from coral reefs and their subsequent transport, as seen in the nudged runs.

Over the MC-Aus region in the free running ensemble, JJA and SON have decreases in $CCN_{70}$ of -1.4 % ($p < 0.1$ and $p > 0.1$ respectively). In MAM, an increase is observed, demonstrating the anomaly in $SO_2$ processes near PNG discussed

earlier, the influence of the free running meteorology and the non-linearity of this system. The violin plots in the far right column of Figure 7 indicate large model spread in the $CCN_{70}$ changes. SON (Figure 7o) has the most ensemble agreement over the MC-Aus region, while little agreement is observed in other seasons. For this reason, from this point only the SON results will be shown and discussed (although the statistics for all seasons can be found in the continuation of Table 1).

Whilst the changes in aerosol discussed in this section are small, it is worth noting that in Fiddes et al. (2018), where all

marine $DMS_w$ was removed, a global decrease of 8 % of all $N_3$ (17 % for Australia) was found. For $CCN_{70}$, a decrease of just 5 % was found globally (8 % for Australia). It is clear from the perturbation of total marine DMS that global DMS contributes only a small amount to the total aerosol number. Hence local differences from coral reef-derived DMS found in this study could be considered to be relatively large.

### 3.5 Direct aerosol radiative effects

Aerosol in the Aitken, accumulation and coarse modes are used by the ACCESS-UKCA radiation scheme to calculate aerosol direct effects. In Figure 8, the SON AOD, clear sky outgoing shortwave radiation at the top of the atmosphere (abbreviated to $SW\uparrow_{\mathrm{TOA,CS}}$) and clear sky incoming shortwave radiation at the surface ($SW\downarrow_{\mathrm{Surf,CS}}$) are shown. The clear sky radiation fields are examined in this Section to allow detection of direct effects from aerosol without the influence of clouds. Changes in the all sky fields are dominated by the convective cloud response, which will be discussed in Section 4.

The spatial response of AOD to changes in aerosol in the nudged simulations (Figure 8b) is broadly similar to the $CCN_{70}$ changes (Figure 7n). Decreasing AOD is found over most of the MC-Aus region, while the area to the west of PNG experiences



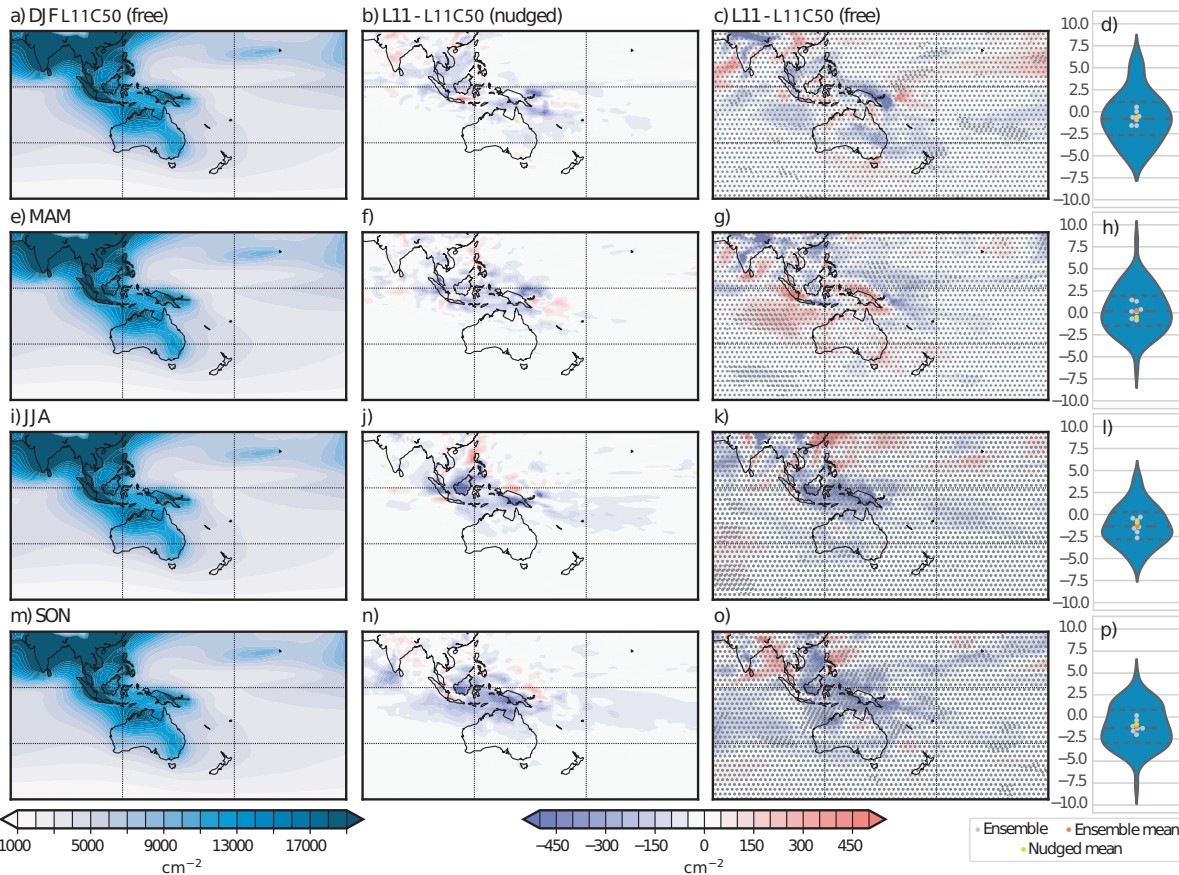

**Figure 7.** The seasonal (DJF - top row, MAM - upper middle row, JJA - lower middle row and SON - bottom row) total column $CCN_{70}$ concentration (in $cm^{-2}$) over the broad MC-Aus region for the free L11C50 simulation (left column); the L11-L11C50 difference for the nudged simulations (middle left column); and the L11-L11C50 difference for the free running ensemble (middle right column) with the model agreement shown by stippling, where at least five of the seven ensemble pairs agree in the sign of the difference. The right column shows violin plots of the average seasonal difference (L11-L11C50) in total column $CCN_{70}$ over the MC-Aus region as a percentage, where all years in the ensemble is shown by the distribution, the dashed lines represent the 25, 50 and 75th percentiles, the grey dots show the mean of each pair of models, the red dots the ensemble average and the yellow the nudged average

an increase. The seemingly amplified aerosol response in the PNG region is due to the increased accumulation mode sulfate mass, and to a lesser degree the increased particle concentration numbers once again associated with the $SO_2$ anomaly discussed earlier. This increase is not believed to be associated with coral reef-derived DMS removal but a function of complex non-linearities and model sensitivities in a polluted region. This anomaly is likely to be dampening the area averages over the MC-Aus region.

**Figure 8.** SON averages over the MC-Aus region for top row: the AOD (unitless and changes in percentage); upper middle row: $SW\uparrow_{TOA,CS}$ ($W\,m^{-2}$); lower middle row: $SW\downarrow_{Surf,CS}$ ($W\,m^{-2}$); and bottom row: column average specific humidity ($g\,kg^{-1}$ and percent). The left column shows the free L11C50 simulation; the middle left column shows the L11-L11C50 difference for the nudged simulations; the middle right column shows the L11-L11C50 difference for the free running ensemble with the model agreement shown by stippling, where at least five of the seven ensemble pairs agree in the sign of the difference; and the right column shows violin plots of the average seasonal differences (L11-L11C50) as a percentage for AOD and specific humidity and in $W\,m^{-2}$ for $SW\uparrow_{TOA,CS}$ and $SW\downarrow_{Surf,CS}$, where all years in the ensemble are shown by the distribution; the dashed lines represent the 25, 50 and 75th percentiles; the grey dots show each pair of models mean, the red dot the ensemble average and the yellow the nudged average





In Figure 8f, the change in nudged $SW\uparrow_{TOA,CS}$ is consistent with the change seen in AOD, indicating that there is a weak reduction in the amount of shortwave radiation being reflected out to space at the top of the atmosphere over significant coral reef regions. This decrease in $SW\uparrow_{TOA,CS}$ suggests more shortwave radiation is passing through the atmosphere and reaching the surface. For the nudged runs, this is broadly true, as shown by the $SW\downarrow_{Surf,CS}$ in Figure 8j. These results indicate that when no interaction with meteorology is allowed, a weak, statistically insignificant direct aerosol effect is associated with certain regions over the MC under clear sky conditions in austral spring.

For the free-running ensemble, however, the result is far less clear, primarily due to interactions with meteorology. The AOD response (Figure 8c) is again broadly consistent with the changes in $CCN_{70}$ (Figure 7o), where over the MC-Aus region a decrease of -1.7% ($p < 0.1$) is found. The $SW\uparrow_{TOA,CS}$ responds as expected with a decrease of $-0.11\,\mathrm{W\,m^{-2}}$ over the MC-Aus region ($p < 0.1$, with considerable model agreement). However, the subsequent $SW\downarrow_{Surf,CS}$ response (Figure 8k) is not as clear. While the SON MC-Aus average suggests a weak, insignificant increase in shortwave radiation reaching the surface, the spatial patterns are inconsistent with the AOD, aerosol fields or the $SW\uparrow_{TOA,CS}$.

In ACCESS-UKCA, the shortwave (defined as wavelengths between 0.2-5 $\mu$m) radiative transfer scheme considers not just the scattering and absorption of energy by aerosol and cloud droplets, but also the absorption of energy by water vapour (Edwards et al., 2013). Water vapour has an effect at wavelengths greater than 0.7 $\mu$m. In Figure 8o, the mean change in water vapour throughout the column is shown, which is more spatially consistent with the free running $SW\downarrow_{Surf,CS}$ response than that of the AOD or $SW\uparrow_{TOA,CS}$. This consistency is particularly clear over the Australian region. An increase in water vapour in the column would suggest more absorption of radiation throughout the column, and hence less energy is received at the surface, as found in the $SW\downarrow_{Surf,CS}$ results (Figure 8k). Upper level warming over the tropics (not shown) is also found and we note that changes in temperatures nearer the surface is limited due to the prescribed SSTs. It is emphasised that this interaction of energy with water vapour is only found in the free ensemble, when meteorology is allowed to vary.

The cause of the change in water vapour is difficult to determine as it is intricately linked to both local and large-scale climate processes. For example, meteorological effects found in this study that could explain the water vapour increase include:

- Warming at upper levels (15-20 km) of the atmosphere over the tropics (not shown), indicating more water vapour can be held in the atmosphere, but could also be a result of increased water vapour

- Increased latent heat flux at the surface over the Australian region (not shown), suggesting increased evaporation

- A general increase in vertical motion found for the Southern Hemisphere tropics-mid latitudes (Figure 9b), aligning neatly with the regions of increased water vapour (Figure 9d). Increased vertical motion is accompanied by decreased southwards transport at upper levels and decreased northwards transport at the surface (Figure 9f), suggesting a weakening of the southern branch of the Hadley Cell.

- Increased high level cloud and convective precipitation found over the MC-Aus region, causing increased high cloud cover, an overall decrease in $SW\downarrow_{Surf}$ and increased convective precipitation.



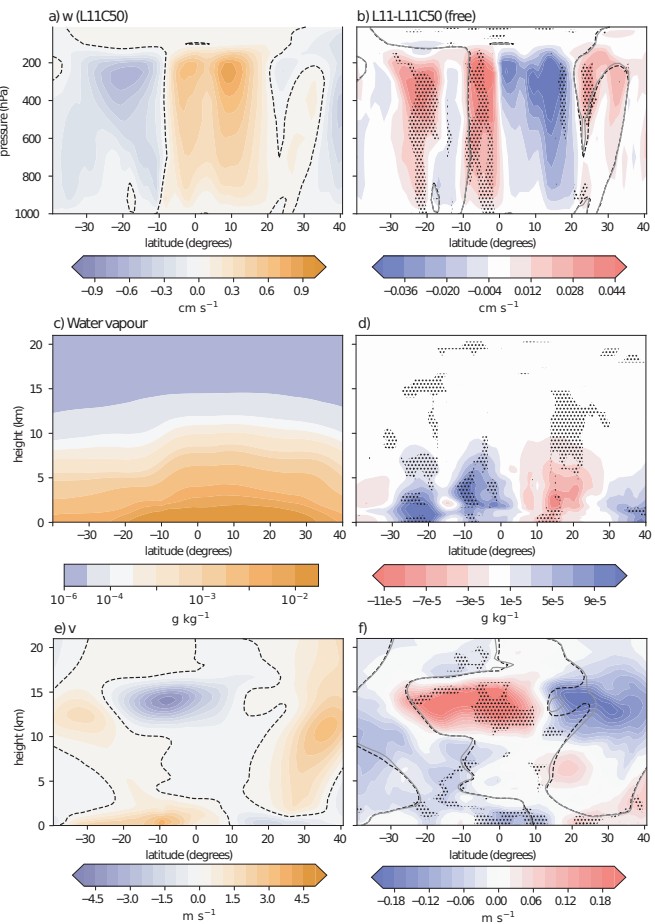

**Figure 9.** The zonal mean vertical profiles over the MC-Aus longitudinal region for vertical velocity (w, top row) in $\mathrm{cm\,s^{-1}}$, water vapour (middle) in $\mathrm{g\,kg^{-1}}$ and the v component of wind (bottom) in $\mathrm{m\,s^{-1}}$) for the free running L11C50 simulation (left column) and the difference in L11-L11C50 the free ensemble (right column), with the model agreement shown by stippling where at least five of the seven ensemble pairs agree in the sign of the difference. For a-b) and (e,f), the black dashed lines in all plots represent the zero contour of the L11C50 field, while in b) and f) the solid grey line indicates the zero contour of the L11 field

It is hypothesised that the small reduction in aerosol resulting from decreased $\mathrm{DMS}_a$ has caused an increase in short wave
radiation passing through the atmospheric column, resulting in warming, which may cause increased evaporation, vertical transport and convective activity, and subsequently increased water vapour. Despite many of the responses described above having good ensemble agreement, low confidence is attributed to this hypothesis for three reasons:





- The top of the atmosphere direct radiative effect that is proposed to initiate these responses is small. Furthermore, the changes in aerosol and AOD that have caused the radiative effect are also small and insignificant. Hence it is unreasonable to suggest such small changes in aerosol could be causing a direct radiative effect.

- Over the MC-Aus region as a whole, little statistical significance is found in these responses, and ensemble agreement can be found in regions that are seemingly not associated with coral reef-derived DMS. Thus, internal model variability cannot be ruled out as the cause for these meteorological responses.

- In Fiddes et al. (2018), where all marine DMS-derived aerosol was removed from the system, decreases in convective activity were found over the tropics, despite those simulations being nudged (nudging allows small differences in meteorology if the forcing is large enough). The change in convective activity induced by removing all DMS is of the opposite sign to that found here.

Therefore, despite a weak yet significant ($p < 0.1$) decline in $\mathrm{SW}{\uparrow}_{\mathrm{TOA,CS}}$, it is concluded that over the MC-Aus region, no robust impact on climate via the direct radiative effect can be confidently detected. This result may be due to averaging over a large area, as the same processes are found over QLD with generally greater statistical significance and ensemble agreement. This is discussed further in Section 4.

### 3.6 Indirect aerosol radiative effects in the large-scale cloud and precipitation scheme

Indirect aerosol effects, such as cloud brightening or lifetime effects, take place as CCN particles activate and become cloud droplets. In ACCESS-UKCA, aerosol activation depends on the size and composition of aerosol as well as the atmospheric supersaturation, which is influenced by the vertical velocity. Thus, the significant increase in smaller size aerosol found in Section 3.2 may have some influence over indirect aerosol effects in certain conditions, although it is noted that larger aerosol have much greater ability to influence indirect effects. Figure 10 shows the responses of cloud droplet number (CDN), cloud liquid water path (LWP) and low cloud fraction (noting that only the cloud microphysics scheme can respond to change in CDN, not the convective scheme). For both the nudged and free simulations, the changes found in CDN are consistent with, though weaker than, the changes observed in the $\mathrm{CCN}_{70}$ fields (noting again the vertical integral of CDN through the column is shown). No statistical significance is attributed to the changes in CDN over the MC-Aus region, although reasonable ensemble agreement is found in Figure 10c. These weak changes suggest that the reductions in CDN are unlikely to have an effect on cloud properties or large-scale precipitation (via the second indirect effect where fewer CDN, given the same availability of liquid water, would increase rainfall).

A small increase in LWP is found on average over the MC-Aus region in the free simulations, although Figure 10g-h shows a large amount of variability in the region and little model agreement. This change in CDN and LWP has had little impact on properties such as low cloud fraction (Figure 8k) or large scale precipitation (not shown), as shown in Table 1, over the MC-Aus region for each season. In addition there is very little model agreement in these responses. For the nudged simulations, constrained meteorology has meant little response of cloud properties is allowed. Subsequently, these results suggest that the loss of coral reef-derived aerosol has little or no impact on climate via the indirect aerosol effects.



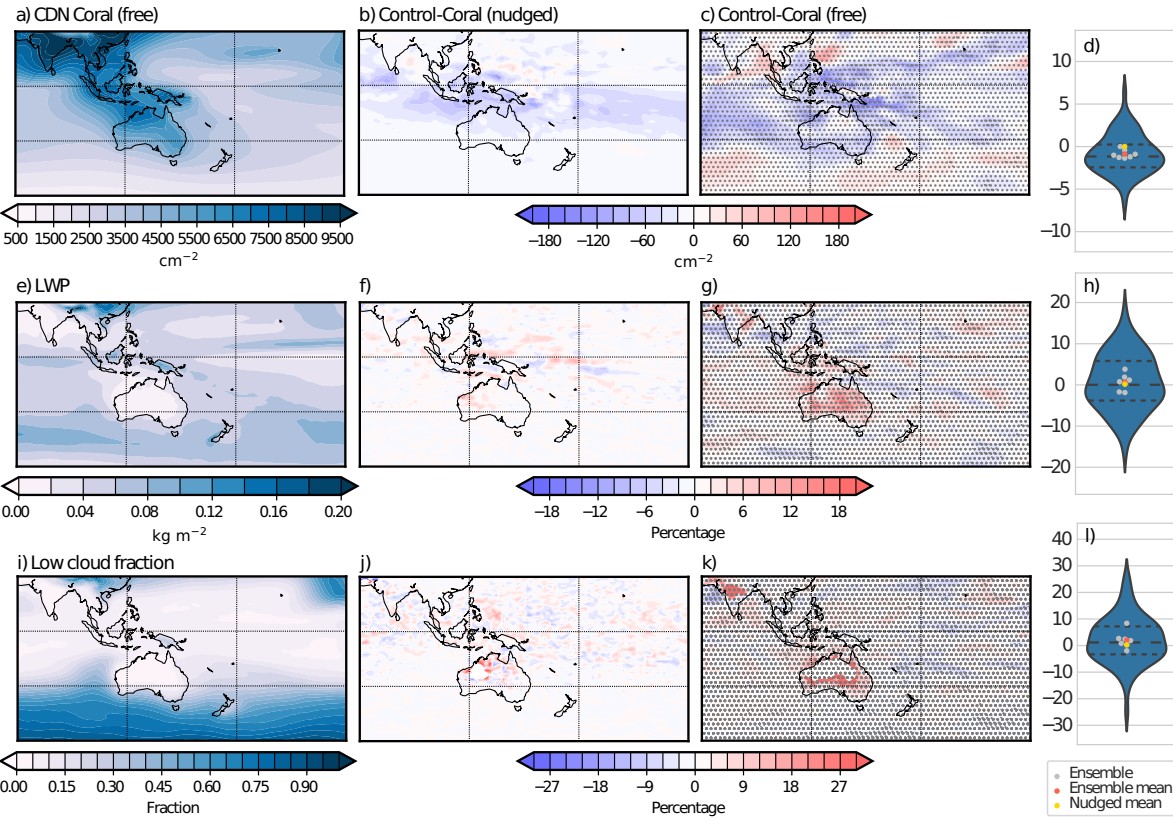

**Figure 10.** As for Figure 8 but for column integral of CDN (top row) in $\mathrm{cm^{-2}}$, LWP (middle row) in $\mathrm{kg\,m^{-2}}$ and percent and low cloud fraction (bottom row) as a fraction and in percent

## 4 Implications for Queensland, Australia

A relatively large increase in precipitation, of 11 %, has been found over QLD in response to removal of DMS produced by coral. Although approximately equivalent increases were found in the large-scale and convective precipitation (10.9 %, $p > 0.1$, and 9.4 %, $p < 0.1$), very little large-scale precipitation occurs in this region, with convective precipitation by far the more important of the two. Furthermore, while CDN concentrations have decreased insignificantly (by 0.5%) and the LWP has increased (7.4% $p < 0.1$) over the QLD region, resulting in more large-scale rainfall, attributing the change in LWP in particular to changes in aerosol is difficult, as discussed in the previous sections.

At this point, it is worth remembering that the convective scheme is not coupled to the aerosol scheme and thus has no knowledge of the removal of coral reef-derived DMS. Hence the increased convective activity must be occurring dynamically, with possible causes discussed in Section 3.5. Specifically, over QLD, a significant decrease in $N_3$ of 5 % is found ($p < 0.05$). Although this does not translate into a meaningful decrease of $CCN_{70}$ (0.5 %), a significant ($p < 0.1$) decline in the AOD is





found of 1.6 %, which has resulted in less SW$\uparrow_{\mathrm{TOA,CS}}$ of -0.16 W m$^{-2}$, ($p < 0.05$). As for the MC-Aus region, the increased radiation allowed through the atmospheric column has been absorbed by increased water vapour (see Figure 8o), resulting in a significant reduction in SW$\downarrow_{\mathrm{Surf}}$ of -0.38 W m$^{-2}$ ($p < 0.1$). While high cloud cover has increased by 7.4 % ($p < 0.1$), which in turn has caused a decrease in SW$\downarrow_{\mathrm{Surf}}$ of -1.61 W m$^{-2}$ ($p < 0.1$), linking these convective responses to the changes in direct aerosol effect at the top of the atmosphere is not able to be done with confidence.

Due to increased high cloud cover, the relatively large change in all sky surface solar radiation is of the opposite sign to what one would expect following the aerosol direct and indirect theories. Further, it is noted that the changes in the all-sky radiation is much larger than that of the clear-sky radiation and indicates that the response in cloud cover is much more important for radiative processes in this region than the direct aerosol effects.

Despite the changes presented in this Section for QLD having greater statistical significance and more ensemble agreement than over the MC-Aus region, low confidence is attached to these results due to an unclear physical mechanism (Section 3.5) and the existence of similar responses elsewhere that are likely to be model noise (not shown). It is emphasised that the meteorological results discussed above are considered to be a response to a direct aerosol radiative effect, only possible when meteorological feedbacks are allowed. Nevertheless, these results were interesting and unexpected, demonstrating a clear example of how non-linear the DMS-climate system is and how important it is to consider the system as a whole, rather than isolating certain aspects.

# 5 Conclusions

This study set out to determine if the loss of coral reef-derived DMS could impact global and regional climate. The ability of coral reefs to produce an aerosol precursor gas has been known for sometime, however the impact of this source of sulfur on the climate has not been quantified until now. On the global scale, coral reefs appear to have little influence on the sulfur budget or global energy balance. At regional scales, however, this work has found some interesting and unexpected effects that highlight the complexity of this system.

The MC-Aus region has been found to have the largest aerosol response to removal of coral reef-derived DMS across the globe. This is unsurprising given that this region has the highest density of coral reefs in the world. Over other coral reef regions, the effects of coral reef-derived DMS are quickly diluted by other influencing factors, such as anthropogenic aerosol sources. Significant decreases in the free running ensemble's small size aerosol (both number and mass) are found over the MC-Aus region when coral reef-derived DMS is removed. For the larger sized aerosols, small, generally insignificant decreases in sulfate mass are found while little change in number is noted. The nudged simulation shows a more consistent small decrease across all aerosol fields and seasons. The decreases in aerosol have cumulatively lead to an insignificant decrease in SON of AOD in the MC-Aus region, and a significant decrease of AOD over QLD. No significant or robust changes were detected in other seasons.

Despite the weak AOD response over the MC-Aus region, a significant reduction in SW$\uparrow_{\mathrm{TOA,CS}}$ of -0.11 W m$^{-2}$ was found for SON in the free running ensemble and attributed to the reduction of aerosol in the region. This decrease is greater than the





equivalent decrease in the nudged simulations. Much smaller (or no) reductions were found in all other seasons. In contrast to the nudged simulations, the free running SON reduction in $SW\uparrow_{\text{TOA,CS}}$ does not directly translate to similar increases in $SW\downarrow_{\text{Surf,CS}}$ as expected. The most likely explanation is that interaction of shortwave radiation with water vapour causes the opposite $SW\downarrow_{\text{Surf,CS}}$ effect, which is itself a result of complex meteorological feedbacks. The response of water vapour could not be confidently attributed to changes in aerosol (as opposed to model variability between ensemble members) and the direct

effects at the top of the atmosphere are found to be weak. For these reasons, this work concludes that no robust direct aerosol effects can be confidently associated with coral reef-derived DMS.

We have found little to no evidence of indirect aerosol effects in any region in response to coral reef DMS. This is unsurprising given the small and insignificant changes found in the CDN and also the results of Fiddes et al. (2018), where few indirect effects were observed outside of the clean marine regions of the Southern Hemisphere mid-latitudes. However, it may also be

a result of the lack of coupling between the convective scheme and aerosol in the model. Coral reefs, by nature, are located in tropical regions that are dominated by convective processes. Hence, it is possible that no indirect effects were found in this work simply because such effects are only allowed to occur in the large-scale scheme, which is not particularly active in the regions of interest.

How convection may interact with changes in aerosol is currently a large source of uncertainty (Tao et al., 2012). Aerosol-

induced convective invigoration theories suggest that with less aerosol, CDN would rain out more quickly, reducing the amount of latent heat release caused by condensation, thus, inhibiting convection. This mechanism has been found in both simulations and observations with respect to increased anthropogenic ultrafine aerosol over the Amazon invigorating deep convection (Fan et al., 2018). However, Nishant et al. (2019) has demonstrated that while satellite observations correlate high aerosol loading to increased convection, the presence of aerosol may not be the cause of connectivity activity due to the co-variation of aerosol-

wind and wind-cloud processes. Fan et al. (2016) summarise that the influence of aerosol on convection is highly dependent on different kinds of convective systems (eg. the trigger mechanism, whether it is a super cell) and the environment (eg. wind shear, cloud base temperature). The studies discussed above and the results of this work indicate that tropical aerosol interaction with climate is far from linear and requires significantly more work to integrate both the convective and large-scale responses in climate modelling.

Interestingly, this study has found some unexpected changes in convective processes, via dynamical mechanism, including a significant increase of 10.9% in convective rainfall over QLD, attributed to suppressed downwards motion in the region. However, low confidence is given to the physical cause (weak direct radiative effects) of the changes in vertical motion in this study.

It should be noted that the weak direct effects discussed above consider clear sky processes only. The changes in convective

processes result in significant changes in high-cloud fraction. Increased cloud cover subsequently has large impacts on the all-sky radiation response (decreased $SW\downarrow_{\text{Surf}}$), overwhelming any clear-sky responses. This again highlights the complexity of the system and issues a warning to the oversimplification of aerosol effects in climate models.

The tool used for this work, ACCESS-UKCA, is able to provide detailed diagnostics of the DMS-climate system; however, important model limitations remain. By conducting a global study, a limitation of this work is the model spatial resolution.





The coarse temporal resolution of the output used in this study (monthly) has made extracting process information difficult, as the averaging applied at these timescales dilutes initial perturbations and the subsequent response. In addition, by allowing the model to run freely, disentangling responses due to coral reef-derived DMS as opposed to internal model variability has been difficult. While an effort to resolve this problem has been made by creating a seven member ensemble (limited by computer resources) and comparing to nudged simulations, there remains a significant likelihood that some of the results presented in

the work are purely from internal model variability. Despite this difficulty, we suggest that future aerosol-climate interaction studies adopt this method of comparing both nudged and free-running simulations in order to separate direct responses from a perturbation from changes occurring via meteorology, and subsequently ensuring a robust physical mechanisms is found explaining the meteorological changes.

Further limitations of this model are the underlying biases in clouds and radiation (Fiddes et al., 2018), and the lack of

representation of some microphysical processes, which can have important effects on how aerosol interact with the climate. In particular we note the single moment cloud microphysics used in this version of ACCESS places a limitation on the modeling of indirect aerosol effects, despite the double moment aerosol scheme. Of note, evaluation performed in Fiddes et al. (2018) showed that ACCESS-UKCA significantly underestimated cloud fractions at all levels, which led to overestimated $SW\uparrow_{TOA}$. The responses found here in cloud and radiation are well within the model uncertainties reported in the literature and are smaller

than the model biases themselves (found in Fiddes et al., 2018). In effect, improved model representation of the climate system may have a larger impact on the energy balance than the small perturbation applied in this work. The sensitivity of global climate models to changes in model structure has been highlighted recently, where new generation models reportedly have a higher climate sensitivity than previously, in part due to improved representation of cloud feedbacks (Zelinka et al., 2020).

The results presented here indicate that although the loss of coral reef-derived DMS may have small impacts on nucleation

and Aitken mode aerosol, the end result is not as straightforward as the aerosol effects theory suggests. This complexity is highlighted by the difference between the nudged simulations and the free ensemble, where in the latter, the impact of changing meteorology in response to a weak direct effect has impacts on the radiation budget often opposing what was expected. These results also demonstrate how interdependent this system is and suggests that not explicitly resolving all aspects of the DMS-climate cycle (chemistry, aerosol, dynamics, clouds, radiation etc) may result in misleading findings. These results further

highlight the costs and benefits of the two modelling methods (free versus nudged simulation) when trying to separate climate responses from climate variability.

This has been the first study to include coral reef-derived DMS in a global climate-chemistry model and to then determine what possible effect the loss of this DMS source may have on climate. It is noted that the amount of $DMS_w$ assumed to be produced by coral reefs in this work (an area weighted 50 nm) is likely larger than reality due to area averaging and the far

more variable nature of DMS production. That said, no robust evidence of indirect effects and weak evidence of a direct effect over just one season (SON) has been found over the MC-Aus region. These results suggest that with smaller estimates of coral reef $DMS_w$, little to no climatic forcing would be found. These results have significant implications for current coral-DMS-climate literature, effectively signalling that DMS produced by coral reefs has little importance for climate via sulfate aerosol radiative forcing. However, we note that we have not evaluated other coral reef links to the climate such as carbon uptake or





ecosystem services. A future study, including simulations conducted with short (hourly) time-scales, is planned to evaluate if coral reef-derived DMS can have a local impact on meteorology, in order to assess the possibility of a bioregulatory feedback system.

Reflecting on the questions posed at the beginning of this study, we are now able to provide the first quantitative evidence that coral reefs likely play little role in regional climate modulation. As coral reefs globally face extinction due to anthropogenic climate change, it is unlikely that the subsequent reduction in precurser aerosols will have a noticeable impact on regional climate. Further modelling studies using future climate scenarios where anthropogenic aerosol dominate, are unlikely to yield different results by the same reasoning. A more interesting question may be what the change in climate would be in a pre-industrial world, with no anthropogenic greenhouse gases and in particular, aerosol pollution. We speculate that under pre-industrial conditions coral reefs may have a greater influence on climate.

*Code and data availability.* The model used for this study is a licensed product of the UK Met Office and is available to specific users under a license agreement. Model simulation output can be made available upon request. Analysis of model output (plotting and statistical analysis) was performed using Python 2.7 and the code can be made available upon request for analysis.

*Author contributions.* SLF completed the model simulations, analysis and write-up presented in this study. MTW provided project guidance, specific model set-up and analysis advice and contributed to the subsequent revisions of this work. RS and TPL provided project guidance, analysis advice and provided comment on the revisions of this paper.

*Competing interests.* The authors have no competing interests to declare

*Acknowledgements.* SLF was supported by the Australian Research Council (ARC) Centre of Excellence for Climate System Science (CE110001028). TL is supported by the Australian Research Council (ARC) Centre of Excellence for Climate Extremes (CE170100023). SLF, RS, were supported by the ARC Discovery Project: Great Barrier Reef as a significant source of climatically relevant aerosol particles (DP150101649) and RS has funding from the ARC Discovery project: : Tackling Atmospheric Chemistry Grand Challenges in the Southern Hemisphere (DP160101598). This research was undertaken with the assistance of resources and services from the National Computational Infrastructure (Project q90 and w40), which is supported by the Australian Government. SLF was supported by the Australian Government Research Training Program Scholarship. SLF would like to thank Georgina Harmer for her help in the design of Figure 1a.





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
