# Peer review of "Coral reef-derived dimethyl sulfide and the climatic impact of the loss of coral reefs"

_Atmospheric Chemistry and Physics, 2020_

## Referee Comment (RC1) · Anonymous Referee #1 · 26 Nov 2020

Dimethyl sulfide (DMS) is the major sulfur source in the remote marine boundary layer. It affects aerosol abundance and cloud formation and growth and thus is very important for the climate. Fiddes et al. conducted model simulations to investigate the impacts of the loss of coral reef, a source of DMS, on the sulfur, aerosol and climate. This work is novel and shows interesting and important results to the community. It could offer useful guidance for making climate change policy. The manuscript is in general well written and scientifically sound. The manuscript can be improved by describing how DMS emission is parameterized (e.g. as a function of wind speed) and how to get the weighted concentration of DMSw (e.g. up to 50 nM). Some minor comments are shown below before acceptance for publication.

1. Line 8: "global flux": is it DMS flux or sulfur flux? Please clarify.

[Figure]

2. Since DMS emission is critical in this work, the introduction section should include how DMSw and DMSa were parameterized in the model in previous studies.

3. Line 29: It is not clear what the "global cooling effect of up to 0.45 C" is relative to.

4. Line 38: Please clarify "when stressed".

5. Line 40: How big is 0.02 Tg/year compared to global DMS flux? Please introduce here.

6. Line 50: Please clarify where the DMS flux is.

7. Line 87: Please show how Lana et al. (2011) calculated DMS emission flux and show more details about the Liss and Merlivat (1986) flux parameterization.

8. Line 97: Please explain how to get the weighted concentration of DMSw.

9. Line 102: Please explain how to get the 50 nM climatology, and provide references.

10. Section 3.1: Please introduce here or in the introduction section how DMS is converted to sulfur dioxide. In addition, please it is useful to show the conversion ratio of "DMS to SO2" in this study and compare that to previous studies.

11. Line 152: Please show the dependence of DMS flux on wind speed.

12. Figure 3: Is the DMSa concentration surface or tropospheric mean? Not clear.

13. Please define the size of nucleation mode, Aitken mode, accumulation mode, and coarse mode aerosols used in the model.

14. Line 178: "decreases by -4.6%" should be "decreases by 4.6%". Also see lines 220-221.

---

## Referee Comment (RC2) · Anonymous Referee #2 · 21 Dec 2020

Summary: The paper presents a first assessment to my knowledge of the impact of coral reef-derived DMS (DMS being an important precursor to sulfur dioxide and subsequent sulfate aerosol formation) on regional climate over the Maritime Continent and Australian regions. The authors implement a new source of DMS derived from coral reefs, in addition to the existing, widely-used DMS climatology of Lana et al. (2011), in the ACCESS-UKCA model and attempt to answer the question of will the loss of coral reef ecosystems have a significant impact on regional climate via associated changes in aerosol-radiation-cloud interactions. The authors conduct both nudged and free-running climate simulations (with and without the coral DMS source applied) to investigate this question and present an in-depth evaluation of the impact of removing the coral source on atmospheric DMS, sulfur dioxide, aerosol properties as well as

top-of-atmosphere and surface radiation and cloud properties. Impacts on DMS, sulfur dioxide and aerosol number concentrations in the nucleation and Aitken size modes are found to be small (all <10%) but statistically significant in some seasons. Impacts on aerosol forcing relevant variables such as AOD and cloud droplet number concentrations are found not to be statistically significant in most instances and changes in the TOA and surface radiative fluxes are very small. The authors subsequently conclude that DMS derived from corals has a very small climatic impact. Overall, I find this paper well-written, well-structured and easy to follow and the figures are all of a very high quality and clear. It is a relevant and interesting topic given the high potential for increasing damage and loss of coral ecosystems and this study on the impacts on aerosols and subsequent climate interactions is novel. Despite the likely negligible impact of this source on aerosol-climate interactions in coral reef regions it is still important to publish such results. I would recommend publication in ACP subject to a few clarifications and minor revisions.

General comments:

My main issue with this study is the use of both nudged and free-running simulations. Nudged simulations are most useful when conducting an evaluation of model against observed variables for a given time and place and are also useful to isolate the aerosol forcing signal in shorter runs than would otherwise be possible in free-running experiments. However, in the latter double-call radiation diagnostics are used to determine both the direct and indirect aerosol effects cleanly. Otherwise, the nudging suppresses the rapid adjustments due to the aerosol perturbation – this is clearly seen for instance in Figure 10. I would therefore argue that nudged experiments aren't appropriate for this current study and see a lot more value in the 7-member free-running ensemble. The authors do highlight how the difference in nudged and free-running experiments highlights the dynamical feedbacks evident in the latter but the analysis carried out on the vertical velocities and water vapour responses lead to the same conclusion so I do question the usefulness of the nudged simulations here and would urge the authors to

do the same.

Specific comments:

DMS sources from coral reefs are reported to be taken from the UNEP-WCMC climatology. Given the relevance of this data source for this study a more detailed description/summary of this dataset is required.

I think it could be useful to break down the annual mean flux in DMS from corals into its seasonal contributions, given much of the analysis of the response is broken down into the seasonal response. Is there a correlation between the seasonality in the source and the response or are there other factors involved?

P7 L170-172 Given that the Woodhouse et al. 2019 study is unsubmitted a brief discussion of the physical mechanism behind the increase in SO2 and in general SO2 sensitivity in this region is required (see also response in accumulation and coarse model aerosol on Section 3.3). Also, what is the role of anthropogenic sources in this region? This is briefly alluded to later (P15 L250) but not discussed in any detail.

Similarly, can the authors comment on the uncertainties in the response due to other aerosol sources in the region?

What is the reason behind SON showing a larger response in CCN than other seasons? No physical mechanism or justification is currently given. Table 1 seems to suggest that MAM also has statistically significant changes of a similar magnitude. Please provide justifiable reasoning behind selecting SON over other seasons for the subsequent analysis.

Technical corrections:

P2 L38 has summarized reports of –> reports

P4 L71 is upon –> on

P4 L82 parameterization –> scheme or model

P4 L86 DMSw –> I don't think this has yet been defined

P5 L108 in this thesis –> in this work or study

P5 L110-112 Is all this really just saying that the 50nM perturbation represents a maximum possible contribution from coral reefs?

P5 L116 hear on –> hereafter

P5 L116 refereed –> referred

P7 L145 free –> free-running

P7 L146 there is no Figure 1c?

P11 L191 removed –> remote

P14 L246 An increase in AOD to the west of PNG is referred to in the text, however I can not see any such increase in fig 8b?

P21 L363 How can the change in OSW be attributed to aerosol if the aerosol changes in themselves are not significant?

P22 L397 oversimplification –> it would probably be more accurate to highlight certain missing interactions / processes here, such as aerosols interactions in the convective plume, rather than just a general oversimplification. Aerosol process representation in models are increasingly complex but this work correctly highlights certain shortcomings of relevance to tropical aerosol-climate interactions.

P23 L401 I'm not sure I agree with all the limitations specified here. Of course resolution is always a limiting factor when it comes to resolving sub-grid scale processes and leads to the need for convective parameterization but there are still tools for pulling out the aerosol signal even at these resolutions and timescales to circumvent the averaging and process extraction issues noted by the authors such as using double-call radiation diagnostics to diagnose the direct and indirect radiative effects as well as using cloud

simulators to determine the effects on clouds.

P23 405-408 As stated above in my main remarks I don't agree with this statement and find a very limited utility of the nudged simulations in this study. The direct responses can be separated from indirect dynamical response through use of double call diagnostics.
* * *

---

## Author Comment (AC1) · 3 Mar 2021

**Response to Reviewer 1**

We would like to thank this Reviewer for their positive comments about our manuscript. As outlined in the line by line response below, we have now included more detailed descriptions of the coral reef  $DMS_w$  climatology and the fluxDMS parameterisation. We believe the changes have strengthened our paper.

**1. Line 8: "global flux": is it DMS flux or sulfur flux? Please clarify.**

We have revised this sentence to:

Line 7: 'A simple representation of coral reef-derived DMS is developed and added to a common DMS surface water climatology, resulting in an additional flux of  $0.3 \,\mathrm{Tg}\,\mathrm{year}^{-1}\,\mathrm{S}$ , or 1.7% of the global sulfur flux'

2. Since DMS emission is critical in this work, the introduction section should include how DMSw and DMSa were parameterized in the model in previous studies.

Our previous paper, which has been identified as a companion paper to this work, details how previous studies parameterise both surface water DMS concentrations and the DMS flux. We have made reference to this in the introduction of this paper:

Line 30: 'Our previous paper, Fiddes et al. (2018), describes these studies, DMS surface water climatologies and flux parameterisations in more detail.'

3. Line 29: It is not clear what the "global cooling effect of up to 0.45 C" is relative to.

We have clarified this sentence to:

Line 29: 'providing global cooling via direct and indirect aerosol effects of up to 0.45 C (Fiddes et al. 2018) when compared to a world in which no marine DMS exists.'

4. Line 38: Please clarify "when stressed".

To avoid confusion, we have removed 'especially when stressed', as the effects of stress on DMS production by corals is discussed in more detail subsequently (Line 50).

**5. Line 40: How big is 0.02 Tg/year compared to global DMS flux? Please introduce here.**

We have revised this paragraph to now include information about the total global DMS flux:

Line 41: 'Jones et al. (2018) also suggest that total emissions from the GBR are equivalent to  $0.02 \text{ Tg year}^{-1}$  of sulfur, noting that total global sulfur flux from DMS is estimated to be between 9- $35 \text{ Tg year}^{-1}$  S (Belviso et al., 2004a; Elliott, 2009; Woodhouse et al., 2010; Tesdal et al., 2016; Fiddes et al. 2018) and that DMS makes up approximately one fifth of the global sulfur budget (Sheng et al. 2015). The Jones et al. (2018) coral reef flux estimation has been made from measurements both over coral reefs and in the GBR lagoon, and also includes an estimate of the additional flux from tropical cyclones.'

**6. Line 50: Please clarify where the DMS flux is.**

We have revised this paragraph to make clear that the coral species used in the laboratory experiments in Hopkins et al. (2016) are found across the Indo-Pacifc. We have also emphasised the fact that these were laboratory experiments.

Line 52: 'Of interest to this study are the findings from Hopkins et al. (2016), where the effect of tidal exposure on three Indo-Pacific coral species were studied in laboratory experiments. From their results, Hopkins et al. (2016) extrapolate a  $flux_{DMS}$  of 9-35  $\mu$ mol m-2 day-1 over coral reefs from Acropora cf.horrida, while an additional 5  $\mu$ mol m-2 day-1 and 8  $\mu$ mol m-2 day-1 can be estimated from two other species in their experiments (*P. cylindrica* and *S. hystrix*.). These estimates are equivalent to a total of 709-1548  $\mu$ g m-2 day-1 of sulfur (if all species are present), and in this work is further extrapolated to global coral reef coverage (approximately 284300 km2), giving 0.074-0.16 Tg year-1 of sulfur. Whilst these extrapolations are highly speculative in terms of artificial laboratory experiments, estimated exposure time, coverage of coral reefs, account for just three Indo-Pacific species of coral and only DMS

produced during tidal stress, the Hopkins et al. (2016) estimations were the first to attempt to quantify the large-scale flux of coral reef-derived DMS.'

**7. Line 87: Please show how Lana et al. (2011) calculated DMS emission flux and show more details about the Liss and Merlivat (1986) flux parameterization.**

We have included several new paragraphs addressing this comment (see Comments 2, 10 and 11), however, we note that we have not included the mathematical derivation of the Liss & Merlivat (1986) parameterisation as this was detailed in our previous Fiddes et al. (2018) paper.

**8. Line 97: Please explain how to get the weighted concentration of DMSw.**

We have included an additional paragraph (below) and Figure (see manuscript - now Figure 2) to address this comment.

Line 130: 'The gridded areal distribution drawn from the coral reef data base, shown in Figure 2, was then used to weight a fixed concentration of  $\text{DMS}_w$  to be added to the Lana et al. (2011)  $\text{DMS}_w$  climatology. For example, the maximum fraction in any gridbox found in Figure 2 is 15.6% in the northern GBR. In this study, we used 50 nM as the fixed concentration (a number of different  $\text{DMS}_w$  concentrations were tested, from 10 nM to 500 nM). Therefore, at this gridbox, with the highest density of coral reefs, the amount of  $\text{DMS}_w$  added to the Lana et al. (2011) climatology as a coral reef source is  $0.156 \times 50 = 7.8 \text{ nM}$ . The choice of 50 nM was somewhat subjective, in part due to the relatively few estimations of large-scale coral reef production of  $\text{DMS}_w$ , as described in Section 1. Nevertheless, below we describe how this choice aligns with observations found in the literature. In addition, we made a conscious choice to create a climatology that represents a plausible maximum  $\text{DMS}_w$  in an attempt to ensure a response to this perturbation.'

9. Line 102: Please explain how to get the 50 nM climatology, and provide references.

We have added an extra explanation of how we chose the weighted addition of  $50 \,\mathrm{nM}$ . Please see comment above.

10. Section 3.1: Please introduce here or in the introduction section how DMS is converted to sulfur dioxide. In addition, please it is useful to show the conversion ratio of "DMS to SO2" in this study and compare that to previous studies.

We have now included a brief description of the DMS oxidative pathways in ACCESS-UKCA in the Methods Section 2.1. We have not, however, included further discussion on these mechanisms as they are well established in the literature and are not the main objective of this study. The model output did not contain DMS to  $SO_2$  budget terms, thus it is not possible to diagnose the conversion ratio of DMS to  $SO_2$ .

Line 112: 'With online chemistry, ACCESS-UKCA includes four key oxidative pathways to convert DMS into  $SO_2$ , which are shown in Table 2.  $SO_2$  can then be further oxidised into  $H_2SO_4$  (Table 2), after which it can contribute to aerosol growth or new particle formation. Description of these processes can be found in Mann et al. (2010).'

**11. Line 152: Please show the dependence of DMS flux on wind speed.**

We have not included the fluxDMS equations in this manuscript as they are detailed in our previous work (Fiddes et al. 2018). We have however included a description of the Liss & Merlivat (1986) dependence on wind speeds.

Line 107: 'In short, the Liss & Merlivat (1986) scheme calculates the  $flux_{DMS}$  under three windinduced sea states representing smooth (10m wind speeds less than  $3.6 \,\mathrm{m\,s^{-1}}$ ) and rough (10m wind speeds between 3.6 and  $13 \,\mathrm{m\,s^{-1}}$ ) gas transfer as well as wave-breaking and bubble bursting (10m wind speeds greater than  $13 \,\mathrm{m\,s^{-1}}$ ).'

12. Figure 3: Is the DMSa concentration surface or tropospheric mean? Not clear.

We have clarified in the Figure 3 caption that it is the  $DMS_a$  surface concentration.

13. Please define the size of nucleation mode, Aitken mode, accumulation mode, and coarse mode aerosols used in the model.

We have added to the Methods Section 2.1 the following paragraph and subsequent table:

Line 89: 'GLOMAP-mode is a two-moment microphysical aerosol scheme that simulates aerosol mass and number distributions across four soluble modes (corresponding to nucleation, Aitken, accumulation and coarse modes) and, in this work, one insoluble mode (Aitken) (Mann et al. 2010, 2012). The size distributions of these modes are shown in Table 1.'

14. Line 178: "decreases by -4.6%" should be "decreases by 4.6%". Also see lines 220-221.

We have updated the text as suggested by the reviewer and also changed other instances of this occurring throughout.

**Response to Reviewer 2**

We would like to thank this Reviewer for their encouraging comments. We have addressed each comment below and made our best efforts to implement the changes suggested or alternately, explained why we did not.

**General Comments**

My main issue with this study is the use of both nudged and free-running simulations. Nudged simulations are most useful when conducting an evaluation of model against observed variables for a given time and place and are also useful to isolate the aerosol forcing signal in shorter runs than would otherwise be possible in free-running experiments. However, in the latter double-call radiation diagnostics are used to determine both the direct and indirect aerosol effects cleanly. Otherwise, the nudging suppresses the rapid adjustments due to the aerosol perturbation – this is clearly seen for instance in Figure 10. I would therefore argue that nudged experiments aren't appropriate for this current study and see a lot more value in the 7-member free-running ensemble. The authors do highlight how the difference in nudged and free-running experiments highlights the dynamical feedbacks evident in the latter but the analysis carried out on the vertical velocities and water vapour responses lead to the same conclusion so I do question the usefulness of the nudged simulations here and would urge the authors to do the same

We agree that the primary advantage of nudged simulations is in model evaluation against observations. Further, we agree that the most important findings from this paper have been drawn from the free running ensemble. While the results presented here with respect to dynamical feedbacks, which was explored via vertical velocities and water vapour, may seem 'self-evident', we point out that these results would not have necessarily been as clear without understanding the differences between the nudged and free running simulations. We wish to clarify that we have not used double-call radiation output in these simulations. We suggest that while the nudged simulations may not provide key results, they do contribute to the interpretation and narrative of this paper. For this reason, we have left these plots and their discussion in the paper. In light of the reviewers comments we have, however, revised parts of the methods and conclusions to make this clear:

Line 159: 'We note that by nudging the model, we limit the model's ability to respond to the DMS flux perturbations. However, nudged simulations, by restricting meteorological feedbacks, give us an indication as to which responses are purely dynamical as opposed to those that may in fact be caused directly by DMS perturbations.'

Line 435: 'Despite this difficulty, we suggest that future aerosol-climate interaction studies use a large-enough ensemble to reduce the signal due to internal variability and that significant analysis of the dynamical aspects of any aerosol response is performed to ensure a robust physical mechanisms is found explaining meteorological changes. Double call radiation diagnostics can also be a useful tool in this instance to diagnose model feedbacks if computer resources permit.'

**Specific comments**

DMS sources from coral reefs are reported to be taken from the UNEP-WCMC climatology. Given the relevance of this data source for this study a more detailed description/summary of this dataset is required.

We have included more detail on the coral reef database and make clear that this database was used to calculate coral reef distribution.

Line 125: 'To determine the amount of  $DMS_w$  added to the Lana et al. 2011 climatology, we first established where coral reefs were located globally. The UNEP-WCMC et al. (2010) global coral reef distribution database was used to calculate the fraction of each ACCESS-UKCA grid box covered by coral reefs. The UNEP-WCMC et al. (2010) database is the most comprehensive global database of warm water coral reefs, 85% of which was drawn from the Millennium Coral Reef Mapping Project, a remote sensing project at spatial resolutions of up to 30m. '

I think it could be useful to break down the annual mean flux in DMS from corals into its seasonal contributions, given much of the analysis of the response is broken down into the seasonal response. Is there a correlation between the seasonality in the source and the response or are there other factors involved?

The coral reef surface water concentrations added in this work do not vary seasonally. Seasonal variation does occur is a result of the monthly Lana et al.(2012) climatology (which is well described in the literature) and seasonal changes in wind fields. We have made note of this in the methods section.

Line 148: 'We note that at this initial stage, the  $DMS_w$  climatology developed here does not vary in time beyond that of the monthly Lana et al. (2012) dataset.'

P7 L170-172 Given that the Woodhouse et al. 2019 study is unsubmitted a brief discussion of the physical mechanism behind the increase in SO2 and in general SO2 sensitivity in this region is required (see also response in accumulation and coarse model aerosol on Section 3.3). Also, what is the role of anthropogenic sources in this region? This is briefly alluded to later (P15 L250) but not discussed in any detail. Similarly, can the authors comment on the uncertainties in the response due to other aerosol sources in the region?

After consideration, the paragraph beginning on line 167 has been deleted. The changes highlighted are minor, and not central to the interpretation of the results subsequently presented. Thus, the inclusion of that paragraph distracts from the core results of the study. Subsequent references to the  $SO_2$  in this region have also been removed.

Aerosol and precursor emissions from the region are diverse, with contributions from biomass burning, anthropogenic, volcanic, and marine sources. A discussion around the contributions made by each of these sources would have to be informed by multiple additional simulations where each emission source is switched off or varied. Such simulations would be interesting, but far beyond what we want to achieve with this manuscript. A follow-on study currently undergoing manuscript preparation uses a regional model to study aspects of some emission sources, but is not yet ready for publication.

What is the reason behind SON showing a larger response in CCN than other seasons? No physical mechanism or justification is currently given. Table 1 seems to suggest that MAM also has statistically significant changes of a similar magnitude. Please provide justifiable reasoning behind selecting SON over other seasons for the subsequent analysis.

The SON season had the most ensemble agreement of all seasons, as noted at the end of Section 3.4. While other seasons had some statistical significance, a lack of ensemble agreement did not give us confidence in these results. In addition, the SON season had the largest dynamical changes, in particular with respect to changes in precipitation over Australia. For these reasons, we moved to just discussing SON from Section 3.5 on-wards, whilst still providing the MC region averages in Tables 3. We have now made this clearer:

Line 261: 'For this reason (and that the subsequent dynamical results to be discussed in Section 3.5 on-wards were the largest in this season), the SON results will be shown and discussed from this point only (although the statistics for all seasons can be found in the continuation of Table 3)'

**Technical corrections**

**P2 L38 has summarized reports of > reports**

We have left this sentence as is, as Jones et al. 2018 is not only reporting new findings, but also summarising a number of previous studies.

P4 L71 is upon > on

Change made as suggested.

P4 L82 parameterization > scheme or model

Change has been made as suggested.

 $P_4 \ L86 \ DMSw > I \ don't \ think \ this \ has \ yet \ been \ defined$

We have clearly defined this as DMS surface water concentrations.

P5 L108 in this thesis > in this work or study

Change has been made as suggested.

P5 L110-112 Is all this really just saying that the 50nM perturbation represents a maximum possible contribution from coral reefs?

Yes. We have clarified this sentence.

Line 137: 'In addition, we made a conscious choice to create a climatology that represents a plausible maximum  $DMS_w$  in an attempt to ensure a response to this perturbation.'

P5 L116 hear on > hereafter

Change has been made as suggested.

 $P5 \ L116 \ referred > referred$

Change has been made as suggested.

P7 L145 free > free-running

Change has been made as suggested.

P7 L146 there is no Figure 1c?

This has been changed to Figure 1b.

P11 L191 removed > remote

Change has been made as suggested.

P14 L246 An increase in AOD to the west of PNG is referred to in the text, however I can not see any such increase in fig 8b?

This should have read east, and has been amended.

P21 L363 How can the change in OSW be attributed to aerosol if the aerosol changes in themselves are not significant?

We have revised the sentence to make clear that we are referring to statistical insignificance:

Line 391: 'Despite the weak AOD response over the MC-Aus region, a significant reduction in  $SW\uparrow_{TOA,CS}$  of  $-0.11 W m^{-2}$  was found for SON in the free-running ensemble and is attributed, in part, to the reduction of aerosol (despite its statistical insignificance) in the region.'

 $P22\ L397\ oversimplification > it\ would\ probably\ be\ more\ accurate\ to\ highlight\ certain\ missing\ inter$  $actions / processes\ here,\ such\ as\ aerosols\ interactions\ in\ the\ convective\ plume,\ rather\ than\ just\ a\ general$  $oversimplification. Aerosol\ process\ representation\ in\ models\ are\ increasingly\ complex\ but\ this\ work\ cor$  $rectly\ highlights\ certain\ shortcomings\ of\ relevance\ to\ tropical\ aerosol-climate\ interactions.$

This sentence was intended to highlight that in broader literature, aerosol effects may be considered, but without detailed knowledge about what can and cannot be simulated (parameterised convective interactions for one). We have amended this sentence to read as:

Line 424: 'This again highlights the complexity of the system and issues a warning to the oversimplification, or perhaps more appropriately, the lack detailed understanding surrounding the current limitations of aerosol effects in climate models, especially as aerosol schemes are more routinely used in

**climate models.'**

P23 L401 I'm not sure I agree with all the limitations specified here. Of course resolution is always a limiting factor when it comes to resolving sub-grid scale processes and leads to the need for convective parameterization but there are still tools for pulling out the aerosol signal even at these resolutions and timescales to circumvent the averaging and process extraction issues noted by the authors such as using double-call radiation diagnostics to diagnose the direct and indirect radiative effects as well as using cloud simulators to determine the effects on clouds.

We have revised this sentence to now read:

Line 429'By conducting a global study, a limitation of this work is the model resolution, both with respect to model time steps (hourly) and output (monthly), as well as spatial resolution. Higher resolution aerosol-cloud interaction studies have been planned with respect to coral reef-derived DMS to address this issue.'

P23 405-408 As stated above in my main remarks I don't agree with this statement and find a very limited utility of the nudged simulations in this study. The direct responses can be separated from indirect dynamical response through use of double call diagnostics.

We have revised this sentence to now read:

Line 435: 'Despite this difficulty, we suggest that future aerosol-climate interaction studies use a large-enough ensemble to reduce the signal due to internal variability and that significant analysis of the dynamical aspects of any aerosol response is performed to ensure a robust physical mechanisms is found explaining meteorological changes. Double call radiation diagnostics can also be a useful tool in this instance to diagnose model feedbacks if computer resources permit.'